# Support Your Local LMs: Redistributing LM Traffic from Cloud to Edge with TrafficBench

## Abstract

The vast majority of large language model (LLM) queries today are processed by frontier models in centralized cloud infrastructure. However, recent advances have produced small language models ($\leq 20B$ parameters) that match or exceed larger models on many tasks while offering superior energy and cost efficiency. To better understand what fraction of inference workloads can be shifted away from cloud to local compute, we present TrafficBench, a comprehensive benchmark for evaluating query routing between local and cloud-deployed LLMs. TrafficBench is comprised of 1M real-world queries derived from ChatGPT user conversations and naturalistic reasoning queries, with evaluations across 10 state-of-the-art (SOTA) models, 4 hardware accelerators, and 8 performance metrics. Using TrafficBench, we address three critical questions: **(1)** what fraction of current inference queries can be handled by small LMs on local accelerators, **(2)** how effectively can modern routing architectures identify these queries, and **(3)** what are the downstream efficiency implications of local routing? Our analysis reveals that 80.7% of TrafficBench queries can be successfully handled by small local models, with coverage varying by domain—exceeding 90% for creative tasks but dropping just below 68% for technical fields. We start by evaluating existing SOTA embedding- and decoder-based routing approaches, finding that they do not push the Pareto frontier beyond individual local models. To enable better routing, we introduce a novel binary variation of decoder-based routing that achieves superior performance ($F1 = 0.851$) when we have access to large training datasets ($> 100K$); we also show that embedding models excel in data-constrained settings ($< 10K$). When deployed over real-world traffic distributions, our decoder-based router reduces energy by 77.1%, compute by 67.1%, and cost by 60.2% versus cloud-only deployment, while maintaining comparable task accuracy. Our longitudinal analysis from 2023-2025 shows a $9.5\times$ improvement in intelligence efficiency (accuracy per watt), with the fraction of locally-serviceable queries increasing from 23.2% to 80.7%, suggesting significant efficiency gains from better routing systems. We release TrafficBench along with a hardware-agnostic profiling harness for measuring model efficiency metrics (e.g., energy utilization), enabling reproducible benchmarking and supporting new research as models and accelerators emerge.

## 1 Introduction

The vast majority of large language model queries today are processed by frontier models deployed in centralized cloud infrastructure. However, recent advances have produced *local LMs*—small LMs that can be run locally ($\leq$20B parameters) such as Qwen3 (Team, 2025), Llama3.1 (Grattafiori et al., 2024), and GPT-OSS (Agarwal et al., 2025)—that match or exceed the performance of larger models on many tasks, while being more energy and cost efficient (Agarwal et al., 2025). Simultaneously, the proliferation of alternative hardware accelerators—from *local accelerators* like the Apple M4 and AMD Ryzen to *cloud accelerators* like the NVIDIA H200 and AMD MI300X—has created diverse deployment options for inference. As a result, users can download and host weights for *multiple* local LMs (i.e., Qwen3-4B, GPT-OSS-20B), toggle between them, and run inference on local accelerators (Kwon et al., 2023; Ollama, Inc., 2024; Zheng et al., 2024b). Despite these developments, LLM queries continue to default to frontier models on cloud accelerators. We ask: *to what extent can current inference workloads be redistributed to LMs running on local accelerators, and what effect does redistributing traffic have on broader performance metrics (e.g. energy, latency, cost, etc.)?*

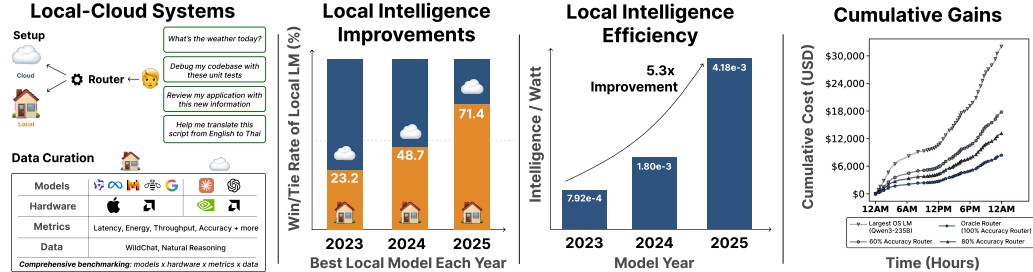

Figure 1: **TRAFFICBENCH Overview.** We present the **TRAFFICBENCH dataset**: *first systematic study of local AI inference efficiency* across models, hardware, and real-world workloads. **(Left)** *Intelligence efficiency is defined as task accuracy per unit of power*, capturing both capabilities delivered and energy consumed. **(Left-Middle)** We conduct *comprehensive performance profiling* across 20+ state-of-the-art local LMs ($\leq 20B$ active parameters), diverse hardware accelerators (APPLE, NVIDIA, AMD), multiple performance metrics, and 1M+ real-world queries spanning chat and reasoning tasks. **(Right-Middle)** *Local LM capabilities are improving rapidly*: win/tie rate versus frontier models increases from 23.2% (2023) to 71.4% (2025)—a $3.1\times$ improvement in accuracy, demonstrating that local models can accurately handle significant portions of single-turn chat and reasoning queries. **(Right)** *Intelligence per watt improves* $5.3\times$ *from 2023–2025*, driven by advances in both model architectures and hardware accelerators, with local accelerators showing $1.5\times$ efficiency headroom compared to enterprise-grade systems.

Towards this, we explore a setup in which queries are routed between a local accelerator running multiple small LMs ($\leq 20B$ parameters) and a cloud accelerator hosting multiple frontier LMs ($\geq 100B$ parameters). We focus on user-generated chat and reasoning queries—the dominant workloads on platforms like Chat-GPT (Chatterji et al., 2025) and Claude.ai (Appel et al., 2025) . Our study is guided by three sub-questions:

**Q1:** **What portion of today's user-generated chat traffic can be effectively handled by LMs on local accelerators today**—and how does this fraction grow as we add more local models or include models from different families? How has this changed over time?

**Q2:** **What is the precision and recall of existing routing architectures—*embedding-based* and *decoder-based*—when identifying queries that can be run locally?** What architecture should we use when we have limited labeled data or are parameter-constrained? Can we train a model that accurately routes each query either to the *smallest* capable local LM or to the cloud when necessary?

**Q3:** **What are the downstream efficiency effects of local LM routing** in terms of cost savings, energy consumption (watts), and compute utilization (FLOPs), when compared to centralized cloud inference? From 2023 onwards, how has *intelligence efficiency* (e.g., accuracy/watt) changed overtime with subsequent generations of models and hardware?

Unfortunately, existing routing benchmarks are inadequate for this study because they: **(1)** lack real *user-generated* queries; **(2)** have not evaluated models across $> 1$ hardware platforms; and **(3)** do not provide efficiency metrics beyond accuracy—such as energy, latency, and compute (Table 1; Section 2).

To fill these gaps, we introduce TRAFFICBENCH, a large-scale, naturalistic routing benchmark (Figure 1). TRAFFICBENCH draws from real ChatGPT queries (captured over one month of user traffic (Deng et al., 2024)) and real-world reasoning queries (Yuan et al., 2025) to create a dataset of 1M queries spanning 22 diverse domains. For each of the 1M queries, TRAFFICBENCH evaluates 10 models, including state-of-the-art local LMs available as of August 2025—QWEN3 (Team, 2025) and GPT-OSS model families (Agarwal et al., 2025)—alongside leading 2023–2024 models (MIXTRAL-8X7B, LLAMA-3.1-8B), over 4 local and cloud accelerators: NVIDIA A100/H200, AMD MI300X, APPLE M4 MAX (Section 3). Beyond accuracy, TRAFFICBENCH reports efficiency measurements—latency, cost, energy, compute, and memory—enabling comprehensive assessments of local LM routing benefits (see **Q3**). To support reproducible and extensible benchmarking as new models, hardware, and tasks emerge, we release the TRAFFICBENCH Profiling Harness—a hardware-agnostic tool for large-scale inference and automated LM evaluation with efficiency metric logging.

To address **Q1**, we study local LM *coverage*: given a set of state-of-the-art (SOTA) local models, what fraction of queries is correctly answered by at least one? Using SOTA local LMs as of August 2025 (QWEN3 4B-14B and GPT-OSS-20B), our analysis on TRAFFICBENCH shows that the smallest local model (QWEN3-4B) achieves 47.9% query coverage, while the largest model (GPT-OSS 20B) achieves 74.8% coverage (Table 2). When considering all four local models together, the joint coverage rises to 80.7%. We observe that joint LM coverage varies by domain—exceeding 90% for creative/social tasks (e.g., Arts & Media) but dropping to 68% in technical fields (e.g., Architecture & Engineering) (Figure 2). Finally, looking at the longitudinal trends of local-cloud systems, **we observe a consistent increase in the ability of local LMs to handle inference workloads with each successive model and hardware generation (2023-2025)** (Table 4). Using the best $\leq 20B$ active-parameter model from each year (2023–2025), small LMs matched frontier performance on 23.2% of total traffic in 2023, 48.7% in 2024, and 74.8% in 2025 (Table 4).

To address **Q2**, we study how much of the local LM coverage (from **Q1**) on TRAFFICBENCH we can capture using modern routing methods. We evaluate the two popular routing approaches—*embedding-based* (Zhang, 2025a; Somerstep et al., 2025; Chen et al., 2024; Guha et al., 2024) and *decoder-based* (Ong et al., 2025)—on their ability to appropriately route queries between models that can be run locally ($\leq 20B$) and models that require cloud accelerators ($> 20B$). Existing approaches for embedding-based and decoder-based routing suffer from low precision/recall scores, preventing them from pushing beyond the Pareto frontier of individual local models (Section 5.2; Figure 3). To enable better routing, we introduce a novel learning approach for decoder-based routing that *decomposes the routing task into individual binary decisions gauging each model's capability*. These decisions are then aggregated before selecting the smallest model capable of answering the query. In terms of absolute performance, **our decoder-based binary router achieves the highest macro F1 scores of 0.875 and 0.825 on the chat and reasoning subsets of TRAFFICBENCH**, outperforming prior decoder-based approaches by +0.136 and +0.177, respectively, and outperforming embedding-based approaches by +0.168 and +0.214, respectively (Figure 3).

To better understand how each of these approaches performs in data- and parameter-constrained environments, we conduct a hyperparameter study sweeping **(1)** model architecture size (160M–8B) and **(2)** training dataset size (10K–1M). We find that **embedding-based routers are better than decoder-based routers in data-constrained ($< 10K$ training data samples) settings** (0.524 vs. 0.423 macro F1 on TRAFFICBENCH) as well as parameter-constrained ($< 1B$) settings (0.491 vs. 0.424 macro F1 on TRAFFICBENCH) (Figure 3; Figure 4). Contrastingly, **decoder-based routers are better than embedding-based routers in data-rich ($> 100K$ training data samples) settings** (0.851 vs. 0.694 macro F1 on TRAFFICBENCH) and parameter-rich ($> 1B$) settings (0.821 vs. 0.705 on TRAFFICBENCH macro F1 on TRAFFICBENCH) (Figure 3; Figure 4). These results suggest that when training data or model capacity is limited, an embedding-based architecture is preferable; however, as routing systems in production continue to gather increasingly large datasets for training, decoder-based architectures are ideal for leveraging the added data.

To address **Q3**, we quantify the efficiency benefits of local LM routing against centralized cloud inference on real-world chat distributions (Wang et al., 2025) over TRAFFICBENCH queries. When applying our decoder-based router we **observe substantial cumulative savings compared to cloud-only deployment: 77.1% lower energy use, 67.1% less compute, and 60.2% cost reduction** (Figure 5). The router achieves Pareto-optimal performance across multiple efficiency metrics (energy, compute, cost), capturing 78.1% of perfect routing efficiency while preserving model quality ($< 5\%$ drop in quality) (Figure 6). When considering these local models *paired with successive generations of accelerators*, we see a **$9.5\times$ increase in intelligence efficiency (accuracy per watt)**—increasing from $3.87 \times 10^{-4}$ in 2023 to $1.22 \times 10^{-3}$ in 2024, reaching $3.67 \times 10^{-3}$ by 2025 (Table 4).

## 2 RELATED WORK

| Benchmark | Real User Queries | Local/Cloud Hardware | Efficiency Metrics | Multi-Model |
|---|---|---|---|---|
| **TRAFFICBENCH** | ✓ | ✓ | ✓ | ✓ |
| RouterBench | ✓ | ✗ | ✗ | ✓ |
| RouteLLM | ✗ | ✗ | ✗ | ✗ |
| RouterEval | ✗ | ✗ | ✗ | ✓ |

Table 1: **TRAFFICBENCH vs. Existing Routing Benchmarks.** Comparison across 4 criteria: (i) contains real user queries, (ii) evaluates over local and cloud hardware platforms, (iii) reports efficiency metrics (e.g., latency, energy, cost, etc.), and (iv) evaluates performance across multiple models.

Our work is inspired by prior routing and inference system studies. See App. A for an extended related work.

**LLM Routing and Benchmarks** A central challenge in local-cloud routing systems is determining which model should handle a given query to maximize efficiency. Prior work spans two main families: embedding-based routers (Zhang, 2025a; Somerstep et al., 2025; Chen et al., 2024) and generative/decoder-based routers (Ong et al., 2025; Chen et al., 2024). RouteLLM demonstrated 85% cost reduction in binary routing (Ong et al., 2024), while ensemble methods like FrugalGPT, RouterDC, and Avengers Pro generalize to multi-model settings (Chen et al., 2023; 2024; Zhang, 2025a;b). Existing benchmarks focus on academic tasks and cost-quality tradeoffs: RouterBench provides 405K curated samples (Hu et al., 2024), RouteLLM evaluates on MMLU and MT-Bench (Ong et al., 2024), and RouterEval compiles 200M records across 8.5K models (Huang et al., 2025), but lack real-world coverage (See Table 1). In contrast, TRAFFICBENCH comprises 1M naturalistic queries, enabling local–cloud routing tradeoff analysis across latency, energy, compute and more on diverse accelerators, and evaluates recent models such as QWEN3 and GPT-OSS (Team, 2025; Agarwal et al., 2025).

**Local-Cloud Inference Systems** Recent work explores collaborative protocols splitting generation between local and cloud LMs. Minions proposes a communication protocol where an on-device LM handles lightweight processing and a frontier, cloud LM performs high-level reasoning (Narayan et al., 2025), speculative decoding employs draft model verification (Miao et al., 2023; Xu et al., 2025), and hybrid systems like SLED and HAT introduce edge-cloud partitioning (Li et al., 2025; Xie et al., 2025). These explore token/layer-level collaboration, whereas we investigate query-level routing.

## 3 TRAFFICBENCH: DATASET AND PROFILING HARNESS

In this section, we provide details on TRAFFICBENCH and the TRAFFICBENCH profiling harness.

### 3.1 TRAFFICBENCH DATASET

**Query Curation** We curate 1M queries across two task categories: chat and reasoning. These categories represent the majority of individual user queries to LLM providers (Zheng et al., 2024a). For chat tasks, we source queries from WILDCHAT (Deng et al., 2024): a dataset of 1M real ChatGPT prompts, spanning 1 month of user traffic. For reasoning tasks, we source queries from NATURALREASONING (Yuan et al., 2025), which provides approximately 1.2 million reasoning-focused queries spanning diverse domains including mathematics, physics, and chemistry. We perform robust data cleaning and filtering (see App. B.1 for details) on each dataset before sampling 500K queries—allocating 450K for training and 50K for evaluation (see Table 2). To provide finer-grained labels to each query beyond "chat" and "reasoning" categories, we use GPT-4O-MINI to further annotate each query with a category from the Anthropic Economic Index (Handa et al., 2025) which maps AI queries to occupations in the U.S. Department of Labor's O*NET. We consider 22 categories in total, spanning "Architecture and Engineering" to "Healthcare Support". The full list appears in App. B.1 (Table 5) along with a breakdown of category representation in the dataset.

**Hardware Accelerators** We run inference on four different AI accelerator systems: the NVIDIA A100 40 GB SXM4 (Ampere) (NVIDIA, 2021), NVIDIA H200 SXM (Hopper) (NVIDIA, 2024), AMD Instinct MI300X (CDNA 3, OAM) (AMD, 2023), and Apple Mac Studio (M4 Max) (Apple, 2024). These systems were chosen because of their different memory capacities (ranging from 40 GB to 192 GB), memory bandwidth (from 546 GB/s to 5.3 TB/s), and power consumption (400W to 750W) (see Table 8 for more details).

**Models** We collect model generations over the QWEN3 (Team, 2025) and GPT-OSS (Agarwal et al., 2025) model families. For the QWEN3 family, we use QWEN3-4B, QWEN3-8B, QWEN3-14B, QWEN3-32B, and QWEN3-235B. For GPT-OSS, we consider the GPT-OSS-20B and GPT-OSS-120B models. For our longitudinal analysis, we evaluate MIXTRAL-8X7B (Jiang et al., 2024) as well as LLAMA3.1-8B (Jiang et al., 2024). For each model, we generate responses across all TRAFFICBENCH queries on each of the 4 hardware backends. Full details of inference hyperparameters can be found in App. B.1.

**Metrics** For each `(query, model, hardware)` triple, we collect labels for accuracy, as well as a wide range of efficiency measurements: latency, throughput, time-to-first-token (TTFT), energy consumption, and more (see Table 7 for a full list of metrics). We use LLM-as-a-judge (see App. B.1 for the respective prompts) to score generated responses against reference answers. For WILDCHAT the reference answers are responses from GEMINI-2.5-PRO, the SOTA closed-source model on LMArena (as of August 2025) (Chiang et al., 2024); for NATURALREASONING, the reference answers are the provided ground truth answers.

| Model | Param. (B) | WildChat Solv. % | NaturalReas. Solv. % | Combined Solv. % |
|---|---|---|---|---|
| *Gemma Models* | | | | |
| Gemma-3-1B | 1 | 33.7% | 32.3% | 33.0% |
| Gemma-3-4B | 4 | 67.1% | 58.9% | 63.0% |
| Gemma-3-12B | 12 | 76.8% | 69.1% | 72.9% |
| *IBM Granite Models* | | | | |
| Granite-4.0-Micro | 1 | 20.3% | 56.4% | 38.3% |
| Granite-4.0-H-Micro | 3 | 27.5% | 62.2% | 44.9% |
| Granite-4.0-H-Tiny | 4 | 18.1% | 27.7% | 22.9% |
| Granite-4.0-H-Small | 8 | 35.6% | 69.4% | 52.5% |
| *Qwen Models* | | | | |
| Qwen3-4B | 4 | 28.8% | 54.5% | 41.7% |
| Qwen3-8B | 8 | 49.0% | 57.2% | 53.1% |
| Qwen3-14B | 14 | 49.7% | 60.2% | 55.0% |
| Qwen3-30B-A3B | 30 | 47.4% | 64.0% | 55.7% |
| Qwen3-32B | 32 | 74.7% | 69.1% | 71.9% |
| Qwen3-235B-A22B | 235 | 100.0% | 69.8% | 84.9% |
| *GPT-OSS Models* | | | | |
| GPT-OSS-20B | 20 | 77.4% | 66.9% | 72.1% |
| GPT-OSS-120B | 120 | 88.9% | 64.9% | 76.9% |
| Combined Coverage (All Models) | N/A | 100.0% | 84.8% | 92.4% |

| Dataset Origin | \|Train\| | \|Test\| |
|---|---|---|
| WildChat | 450K | 50K |
| NaturalReasoning | 450K | 50K |

Table 2: **TRAFFICBENCH Overview. (Left)** Dataset composition with split sizes. **(Right)** Model performance comparison. Solvability (*Solv.*) is the percentage of problems each model can solve correctly.

## 3.2 TRAFFICBENCH PROFILING HARNESS

We develop an end-to-end, cross-platform profiling harness for inference workloads, ensuring reproducible results and support for new models, tasks, and hardware backends. It consists of three core components: distributed multi-GPU inference, response evaluation, and system-level telemetry collection. The harness supports three hardware backends: NVIDIA, macOS (Apple Silicon), and AMD. Given a dataset, model, and hardware backend, it orchestrates inference over all input queries, evaluates outputs (via exact match or LLM-as-a-judge), and records detailed telemetry—latency, throughput, time-to-first-token (TTFT), energy consumption, and more (see Table 7). Telemetry is collected via vendor APIs, synchronized at nanosecond resolution, and normalized (watts, joules, megabytes). On NVIDIA systems, we use NVML to retrieve per-device power, energy, memory usage, and temperature. On macOS, we extract GPU power data from `powermetrics` and compute energy via numerical integration. On AMD systems, we query ROCm SMI for power, temperature, and VRAM usage, again integrating power over time. Full details provided in App. B.1.

## 4 ROUTING METHODOLOGIES

We examine two fundamental approaches to model routing and apply them to the local-cloud routing setting: *embedding-based* and *decoder-based routing*. Both methods aim to route each query to the smallest model that can answer it correctly, balancing workloads between local and cloud models.

### 4.1 PRELIMINARIES

**Problem Definition** Let $q \in Q$ be an input text query, and let $\mathcal{M} = \mathcal{M}_{\text{local}} \cup \mathcal{M}_{\text{cloud}}$ be the set of available language models, ordered by parameter count, where:

- $\mathcal{M}_{\text{local}} = \{M_{4B}, M_{8B}, M_{14B}, M_{20B}\}$ represents small models deployable on local accelerators
- $\mathcal{M}_{\text{cloud}} = \{M_{100B+}\}$ represents frontier models requiring cloud infrastructure

In the local-cloud routing setting, the routing problem is to find the optimal model $M^*$ that minimizes model size while maintaining answer correctness for a given query $q$:

$$M^* = \operatorname*{argmin}_{M_i \in \mathcal{M}} \operatorname{size}(M_i) \quad \text{s.t.} \quad c(q, M_i) = 1 \tag{1}$$

where $c(q, M_i) \in \{0, 1\}$ indicates whether model $M_i$ produces an accurate answer for query $q$.

For a given $q$, we wish to learn $r : Q \rightarrow \mathcal{M}$, where $r$ solves the optimization problem in Eq. 1.

**Evaluation Metrics** We evaluate routing performance using four complementary metrics used in previous literature (Barrak et al., 2025; Srivatsa et al., 2024). For models that receive no routed queries, we define

precision as undefined and exclude them from aggregated metrics. First, we compute precision and recall for each model $M_i \in \mathcal{M}$ (where $r(q)$ denotes the routed model for query $q$):

$$\text{Precision}_i = \frac{|\{q : r(q) = M_i \wedge c(q, M_i)\}|}{|\{q : r(q) = M_i\}|}, \quad \text{Recall}_i = \frac{|\{q : r(q) = M_i \wedge c(q, M_i)\}|}{|\{q : c(q, M_i)\}|}, \quad \text{F1}_i = \frac{2 \cdot \text{Precision}_i \cdot \text{Recall}_i}{\text{Precision}_i + \text{Recall}_i}$$

We then aggregate these per-model metrics to obtain router-level performance:

$$\text{Precision} = \frac{1}{|\mathcal{M}|} \sum_{M_i \in \mathcal{M}} \text{Precision}_i, \quad \text{Recall} = \frac{1}{|\mathcal{M}|} \sum_{M_i \in \mathcal{M}} \text{Recall}_i, \quad \text{Macro F1} = \frac{1}{|\mathcal{M}|} \sum_{M_i \in \mathcal{M}} \text{F1}_i$$

Finally, *Routing Accuracy* is the overall fraction of queries routed to models that answer them correctly, where $M_{\text{routed}})$ denotes the model selected by the routing function $r(q)$ for query $q$:

$$\text{Routing Accuracy} = \frac{1}{|Q_{\text{test}}|} \sum_{q \in Q_{\text{test}}} \mathbb{1}[c(q, M_{\text{routed}})]$$

## 4.2 EMBEDDING-BASED ROUTING

Embedding-based routers compress queries into fixed vector representations and use these embeddings to determine model selection. We examine two variants: *k-NN with frozen embeddings* (Zhang, 2025a;a; Chen et al., 2024) and *finetuned embeddings with classifier* (Somerstep et al., 2025).

**k-NN with Frozen Embeddings** This approach uses a pre-trained encoder to embed queries, then performs k-nearest neighbor (k-NN) search to find similar queries in the training set. Given a query $q$, we:

1. Compute embedding $e_q = \text{Encoder}(q) \in \mathbb{R}^d$ using a frozen pre-trained encoder
2. Identify the $k$ nearest training queries $\mathcal{N}_k(q)$ using cosine similarity
3. Route to the most frequently occurring model $M_{\text{routed}}$ among these neighbors

We evaluate this approach across $k \in \{1, 5, 10, 50, 100\}$ to understand the impact of neighborhood size on routing decisions, with ties broken by selecting the smallest model. On TRAFFICBENCH, we find $k = 100$ to be the most effective configuration and use it in Section 5 (see App. C for k-NN hyperparameter sweep).

**Finetuned Embeddings with Classifier** This approach jointly trains a multi-layer perceptron (MLP) classifier and the encoder embeddings to directly predict the optimal model. Given query $q$, we:

1. Encode the query text using a pre-trained encoder (initialized from Sentence-BERT)
2. Pass the embedding through an MLP classifier to obtain model scores
3. Apply softmax to obtain probabilities $P(M_i|q)$ for each model $M_i \in \mathcal{M}$
4. Route to model $M_{\text{routed}} = \text{argmax}_{M_i \in \mathcal{M}} P(M_i|q)$

## 4.3 DECODER-BASED ROUTING

Decoder-based routers leverage autoregressive language models to process queries token-by-token. We examine two formulations: *multi-class classification* (Ong et al., 2025) and *binary classification*, a novel approach we introduce for decoder-based routing.

**Multi-Class Classification** The *multi-class decoder* frames routing as a text generation task where the model directly outputs the name of the optimal model. Given query $q$, we:

1. Process the full query text autoregressively
2. Compute log-likelihood $\ell_i = \log P(M_i|q)$ for each model $M_i \in \mathcal{M}$
3. Apply confidence threshold $\tau$ to filter viable models: $\mathcal{M}_{viable} = \{M_i : \ell_i > \tau\}$
4. Route to the smallest model in $\mathcal{M}_{viable}$

**Binary Classification** This approach reformulates routing as a binary classification task. For each query-model pair $(q, M_i)$, the *binary decoder* predicts whether $M_i$ can successfully answer $q$:

1. Evaluate $P(\text{"Yes"}|q, M_i)$ for each model $M_i \in \mathcal{M}$
2. Compute log-likelihood scores $\ell_i = \log P(\text{"Yes"}|q, M_i)$
3. Apply confidence threshold $\tau$ to filter viable models: $\mathcal{M}_{viable} = \{M_i : \ell_i > \tau\}$
4. Route to the smallest model in $\mathcal{M}_{viable}$

### 4.4 Training and Evaluation Procedure

All experiments utilize the TrafficBench train/test split. We finetune both decoder and embedding routers using cross-entropy loss with the Adam optimizer (Kingma & Ba, 2014), a $5e-6$ learning rate, batch size of 64, linear warmup, and cosine decay. See App. C for hyperparameter search details.

## 5 TrafficBench Experiments

We organize our experiments into three groups, aligned with the research questions **Q1–Q3** from the introduction. First, addressing **Q1** (Section 5.1), we use TrafficBench to quantify the share of queries that local LMs can handle, and how this fraction scales with model count, architectural diversity, domains, and successive LM generations (Tables 3 and 4; Figure 2). Second, addressing **Q2**, we evaluate embedding-based and decoder-based routing approaches in Section 5.2, comparing their precision, recall, macro F1, and data and parameter efficiency, while Section 5.3 analyzes router generalization to out-of-distribution queries and scalability as the number of candidate models grows (Figures 3 and 4). Finally, for **Q3** (Section 5.4), we leverage efficiency metrics to quantify the downstream gains—across energy, compute, cost, and latency, achieved by routing over local LMs instead of defaulting to frontier cloud models (Figures 5 and 6). Additional ablations (e.g., quantization, judge reliability, router overhead, and multi-turn routing) are grouped in App. D.

### 5.1 Local vs. Cloud Traffic Distribution Analysis

| Local LMs | Maximum Parameters (B) | WildChat Routing % | NaturalReas. Routing % | Combined Routing % |
|---|---|---|---|---|
| Qwen3 4-14B | 14 | 56.1% | 68.6% | 62.3% |
| GPT OSS 20B | 20 | 77.4% | 66.9% | 72.1% |
| IBM Granite (All) | 8 | 39.8% | 79.1% | 59.5% |
| Gemma3 1B-12B | 12 | 85.2% | 76.6% | 80.9% |
| **Best Local LM per Query** | N/A | 100% | 84.8% | 92.4% |

Table 3: **Cumulative Routing Coverage on TrafficBench**. Coverage improvement as more local models are added. Routing % indicates the percentage of problems solvable by at least one model.

Under ideal routing conditions—where each query is assigned to the smallest capable model—a suite of SOTA local LMs from 2025 (Qwen3 4B–14B and GPT-OSS-20B) can correctly answer 80.7% of queries without relying on cloud models (Table 3). This distribution varies across datasets, with WildChat achieving 98.4% local LM coverage compared to NaturalReasoning's 63.0%, demonstrating that conversational queries are easier to route to local LMs than reasoning queries (Table 3).

**Scaling Model Count & Diversity** Increasing local model options demonstrates substantial gains. Using only the 4B model enables 47.9% local routing coverage (Table 2). Expanding to four models (4B, 8B, 14B, and 20B) increases coverage to 80.7% (Table 3), a 32.8 percentage point gain. The largest gain occurs when adding the 8B model (+13.8 pp), with subsequent additions providing progressively smaller but meaningful improvements of +5.2 pp and +5.6 pp respectively. Model family diversity yields additional improvements beyond parameter scaling alone. Expanding from Qwen3-only to include both Qwen3 and GPT-OSS models boosts local routing from 64.9% to 80.7%, a +15.8 point gain.(Table 3).

**Longitudinal Analysis** Table 4 tracks the increasing viability of local inference from 2023 to 2025. In 2023, the best available local LM (Mixtral-8x7B) could handle 23.2% of TrafficBench queries. By 2024, Llama-3.1-8B increased this to 48.7%, and in 2025, GPT-OSS-20B reached 78.6%.

**Domain Level Performance** Domain-specific performance reveals variation in local LM coverage, ranging from 93.6% in Arts & Media to only 53.2% in Architecture & Engineering (Figure 2). We find a 27.4 percentage point (pp) gap in local LM coverage between WildChat (93.6%) and NaturalReasoning (66.2%), demonstrated that local LM coverage on reasoning tasks is less (Table 6). Model family analysis shows that combining Qwen3 and GPT-OSS improves local LM coverage across domains, with an average gain of +13.9pp. Qwen3 performs best in educational and creative tasks (up to +4.0%), while GPT-OSS excels in service-oriented areas—e.g., a 14.7% advantage in *Food Preparation and Serving*. The largest combined improvements are seen in healthcare: *Healthcare Support* and *Healthcare Practitioners* improve by +13.4% and +13.3%, respectively, when both families are used.

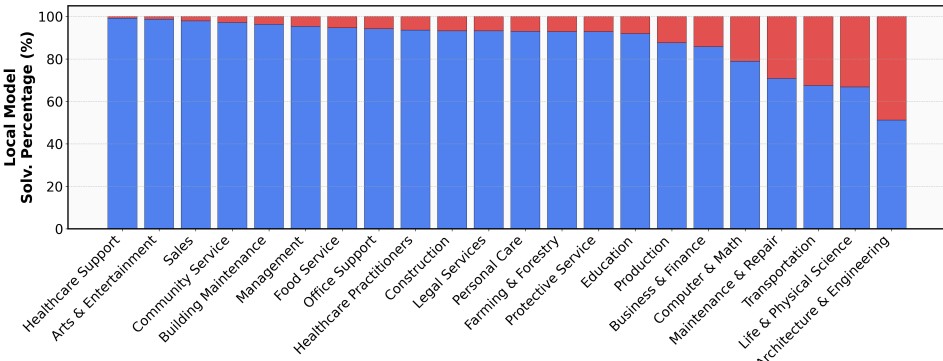

Figure 2: **Local vs. Cloud Traffic Share by Domain**. Stacked bars show the fraction of TRAFFICBENCH queries handled by local LMs ($< 20B$; blue) versus those routed to frontier models in the cloud (red), computed per economic index domain (Appel et al., 2025)

## 5.2   ROUTING APPROACH COMPARISON

Among the 4 routing strategies (Section 4), our binary decoder achieves the best F1—0.875 on WILDCHAT and 0.827 on NATURALREASONING —when trained on 1M queries (Figure 3). This outperforms the next-best (multi-class decoder) by +0.136 and +0.177 F1, respectively. The gain comes from framing routing as independent binary decisions per model, rather than a single multi-class choice (Section 4.3).

**Data Efficiency** Figure 3 highlights distinct data efficiency patterns across routing paradigms. In the low-data regime ($< 10K$ samples), embedding-based methods outperform decoder-based ones by 0.101. This trend reverses beyond 100K samples, where decoder-based approaches surpass embeddings by 0.157. Our binary decoder shows steady gains from 1K to 1M queries, while embedding-based methods saturate early—reaching 90% of their final performance by 20K samples. On WILDCHAT, embedding approaches achieve F1 scores of 0.639–0.671; on NATURALREASONING, 0.585–0.614. The finetuned embedding variant outperforms frozen k-NN by 0.047 macro F1 points on average. Both embedding methods exhibit early plateauing in their learning curves, stabilizing with limited training data.

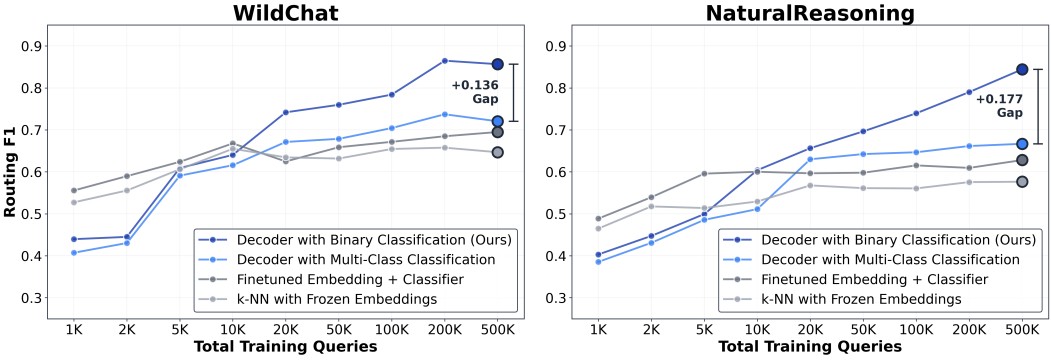

Figure 3: **Learning Curves for Routing Strategies.** Performance of four routing methods on WILDCHAT (left) and NATURALREASONING (right) vs. training size. Binary decoder routing leads throughout, with gains of +0.136 (F1) on WILDCHAT and +0.177 on NATURALREASONING. Embedding methods perform best under 10K queries but plateau early ($\sim$90% of final F1 by 20K). Decoder approaches, especially binary, continue improving up to 1M queries. All results use held-out test sets.

**Parameter Efficiency** Our binary decoder scales strongly with model size, improving from 0.425 F1 at 0.6B to 0.851 at 8B—a gain of 0.426 (Figure 4, left). In contrast, the finetuned embedding classifier improves by only 0.204 (from 0.487 to 0.691). Below 1B parameters, embeddings outperform decoders by 0.062 F1, but this trend reverses at scale: at 8B, decoders lead by 0.172. This demonstrates that embedding routers are efficient at small scales, while our binary decoder router scales better with size.

## 5.3 GENERALIZATION AND SCALABILITY

We evaluate the router's performance along two axes: its ability to generalize to out-of-distribution domains, and its scalability when routing across an increasing number of models (2 to 8).

**Scaling Models** When scaling from 2 to 8 routing targets, our binary decoder router shows minimal F1 degradation ($-0.048$, from $0.911$ to $0.863$), while embedding-based approaches drop significantly ($-0.241$, from $0.892$ to $0.651$) (Figure 4, middle). Our decoder's binary formulation—assessing each model independently—avoids the combinatorial explosion of decision boundaries in multi-class setups. At 8 models, it maintains a clear advantage ($0.863$ vs. $0.651$). These trends suggest that embedding-based approaches are not applicable in settings where the number of routing targets $>2$.

**Cross-Dataset Generalization** When extending to new query distributions—SUPERGPQA (Du et al., 2025) benchmark (Figure 4, right), which tests out-of-distribution routing performance—our binary decoder router (trained on TRAFFICBENCH) achieves $56.7\%$ accuracy compared to $45.3\%$ for an in-domain multi-class decoder router baseline. Moreover, it performs within $6.2\%$ of the cloud LM (QWEN3-235B) performance. This suggests that the binary formulation for our decoder-based router provides better generalization compared to the next best approach (i.e. multi-class decoder-based routers).

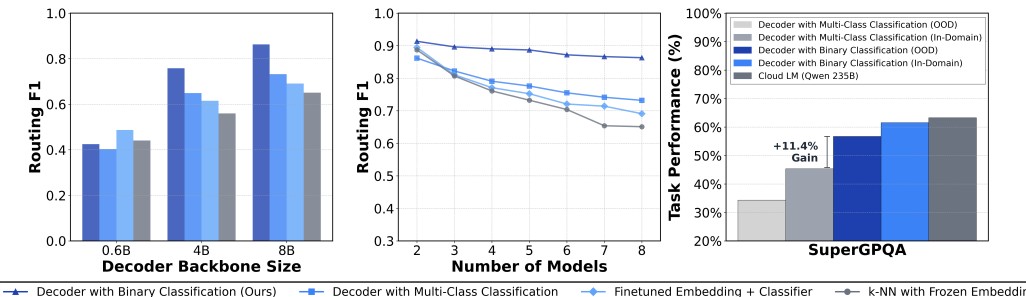

Figure 4: **Parameter Efficiency and Generalizability of Routers**. Analysis of routing performance across 3 dimensions: **(Left)** Scaling backbone size from 0.6B to 8B yields larger F1 gains for binary decoders ($+0.426$) than for finetuned embeddings ($+0.204$). **(Center)** Increasing routing targets from 2 to 8 causes smaller F1 degradation for binary decoders ($-0.048$) than embeddings ($-0.241$). **(Right)** On SuperGPQA, binary decoders generalize better out-of-distribution ($56.7\%$ vs. $45.3\%$ for multi-class decoder).

## 5.4 REAL-WORLD EFFICIENCY AND DEPLOYMENT IMPACT

Using TRAFFICBENCH, we measure the impact of routing real-world LLM queries to local models (QWEN3 4B–14B, GPT-OSS-20B) running on an Apple M4 Max local accelerator, compared to sending all queries to a large cloud model (QWEN3-235B) running on a NVIDIA H200 cloud accelerator. We evaluate this across three axes: **(1)** cumulative efficiency gains from routing under real traffic distributions, including energy, compute, and cost reductions, **(2)** accuracy-efficiency tradeoffs achieved by routing under confidence thresholds, and **(3)** intelligence efficiency trends from 2023–2025 to show how improvements in local models and hardware accelerators together drive increasing efficiency gains.

**Cumulative Efficiency Gains** Deploying our binary decoder router on real-world query distributions (Wang et al., 2025) over TRAFFICBENCH queries yields cumulative savings of $77.1\%$ in energy consumption, $67.1\%$ in compute utilization, and $60.2\%$ in operational costs over a 24-hour period serving 80.2 million requests, compared to routing all queries to frontier models (Figure 5). These efficiency gains emerge from routing $75.1\%$ of queries to local models while maintaining comparable task accuracy, achieving $78.1\%$ of the theoretical optimal (oracle) routing efficiency.

**Accuracy-Efficiency Tradeoffs** Figure 6 demonstrates that our binary decoder router achieves Pareto-optimal performance across 3 efficiency metrics: latency, energy, and cost. Routing to local models reduces average latency from 8.0s (QWEN3-235B on a H200) to 2.1s, while maintaining $93.4\%$ task accuracy. Energy consumption scales superlinearly with model size: the 235B model consumes 3,000J per query compared to 1100J, on average—a $2\times$ reduction. Operational costs follow suit: \$0.0014 per query for the 235B model versus \$0.00013 for local models, retaining $85\%$ of cloud-only accuracy (see Table 12 for pricing

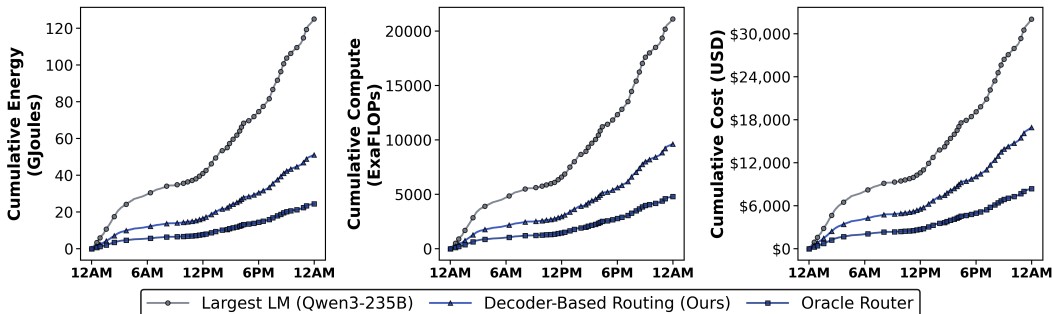

Figure 5: **Energy, Compute, and Capital Gains from Model Routing**. Cumulative resource consumption over 24 hours and 80.2M LLM queries (Wang et al., 2025). Using our local-cloud router between 4 small LMs on Apple M4 Max and QWEN3-235B on an H200 yields substantial savings: 77.1% in energy, 67.1% in compute, and 60.2% in cost compared to naively routing every query to QWEN3-235B, capturing 64–81% of the gains achievable by the theoretical best-case (Oracle Router).

details). For a single confidence threshold setting, the router balances these tradeoffs, achieving 91.2% of frontier model accuracy while reducing average latency by 73.4%, energy by 68.1%, and costs by 47.2%.

**Intelligence Efficiency** When paired with advances in accelerators—from NVIDIA H100 (2023) to Blackwell (2025) —intelligence efficiency (accuracy per watt) has increased from $3.87 \times 10^{-4}$ to $3.67 \times 10^{-3}$, representing a $9.5\times$ improvement (Table 4). The convergence of more capable local LMs and more efficient accelerators suggests that local routing viability will continue expanding.

| Year | SOTA Local Model | SOTA Accelerator | Local % of TRAFFICBENCH | Accuracy per Watt | YoY Efficiency Gain |
|------|------------------|------------------|-------------------------|-------------------|---------------------|
| 2023 | Mixtral-8x7B-v0.1 | NVIDIA H100 | 23.2% | $3.87 \times 10^{-4}$ | — |
| 2024 | Llama-3.1-8B-Instruct | NVIDIA H200 | 48.7% | $1.22 \times 10^{-3}$ | 3.15× |
| 2025 | GPT-OSS-20B | NVIDIA Blackwell | 74.8% | $3.67 \times 10^{-3}$ | 3.01× |

Table 4: **Longitudinal Change in Local Model Capabilities from 2023 to 2025**: Accuracy per watt has improved nearly $9.5\times$ in two years, driven by advances in both model architectures (from MIXTRAL-8X7B to GPT-OSS-20B) and accelerator hardware (from NVIDIA H100 to Blackwell).

## 6 CONCLUSION

We introduce TRAFFICBENCH, a comprehensive benchmark of 1M real-world queries for evaluating routing between local and cloud-deployed LMs. Our analysis reveals that 80.7% of current LLM inference workloads can be successfully handled by small LMs on local accelerators, varying from over 90% for creative tasks to 68% for technical domains. When comparing our binary decoder router to embedding-based routers, we observe that our router achieves superior performance ($F1 = 0.851$) in high-resource settings ($>100$K samples), while embedding-based methods excel when resources are constrained ($<10$K samples). Deployed on real-world traffic distributions, our router achieves 77.1% energy reduction, 67.1% compute savings, and 60.2% cost reduction compared to cloud-only deployment, while maintaining comparable task accuracy. Longitudinal analysis shows the fraction of locally-serviceable queries increased from 23.2% in 2023 to 80.7% in 2025, with intelligence efficiency (accuracy per watt) improving by $9.5\times$. Future work should explore adaptive routing policies, collaborative local-cloud protocols, and domain-specific model optimization to further increase local coverage.

## ETHICS STATEMENT

This work introduces TRAFFICBENCH, a benchmark for evaluating LLM query routing between local and cloud deployments. Our research uses publicly available datasets (WildChat and NaturalReasoning) that contain anonymized user queries. We do not introduce new risks beyond those inherent in the underlying language models being routed. The primary ethical consideration of our work is its positive environmental impact: by demonstrating that 80.7% of queries can be handled by smaller local models, we show a path toward reducing the energy consumption of AI systems by up to 77.1% (Section 5).

## REPRODUCIBILITY STATEMENT

We are committed to ensuring the reproducibility of our results and provide extensive resources to support replication:

**Dataset Release:** The complete TRAFFICBENCH dataset containing 1M queries with performance annotations across 23 domains will be made publicly available upon publication. This includes train/test splits, domain labels, and model correctness annotations.

**Profiling Harness:** We plan to release the TRAFFICBENCH Profiling Harness, a hardware-agnostic tool for large-scale inference and automated LM evaluation with comprehensive telemetry logging. The harness supports NVIDIA, AMD, and Apple Silicon platforms, with documented APIs for extending to new hardware.

**Implementation Details:** All hyperparameters for routing architectures are specified in Section 4 and Appendix C, including learning rates ($5e-6$), batch sizes (64), training epochs ($5-10$), and architecture configurations. We use consistent inference parameters across all experiments: temperature=0.6, top-p=0.95, top-k=20, with a 4096-token output limit.

**Hardware Specifications:** Complete hardware details are provided in Table 8, including specific GPU models (`NVIDIA A100/H200`, `AMD MI300X`, `Apple M4 Max`), memory configurations, and power specifications.

**Model Weights:** We evaluate publicly available models including `Qwen3` (4B-235B variants) and `GPT-OSS` (20B, 120B). Model-specific inference configurations, including repetition penalty (1.1) and length penalty (1.0) for Qwen models, are documented in Appendix B.

**Computational Requirements:** Training the decoder-based router (8B parameters) requires approximately 48 GPU-hours on an NVIDIA H100. Generating model outputs and computing performance metrics for the complete TRAFFICBENCH dataset requires approximately 1,200 GPU-hours across all combinations of datasets (WildChat, NaturalReasoning), models (Qwen3 variants, GPT-OSS), and hardware platforms (`NVIDIA A100/H200`, `AMD MI300X`, `Apple M4 Max`). These requirements can be reduced by evaluating fewer model-hardware combinations or using subsampled datasets for initial experiments.

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

## A    EXTENDED RELATED WORKS

Below we provide an extended treatment of related works.

**LLM Routing**    A central challenge in local-cloud routing systems is determining which model should handle a given query so as to maximize efficiency. Prior work spans a broad design space, but much of it can be organized around two families of approaches: embedding-based routers (Zhang, 2025a; Somerstep et al., 2025; Chen et al., 2024) and generative/decoder-based routers (Ong et al., 2025). Embedding-based methods rely on encoding queries (and sometimes models) into a vector space and then applying similarity search or lightweight classification. Early work largely adopted binary routing, where queries are directed between just two models. For example, RouteLLM (Ong et al., 2024) demonstrated that simple supervised classification can yield up to 85% cost reduction while maintaining GPT-4–level performance, but this setting was restricted to two-model scenarios. More recent systems generalize routing to multi-model settings: ensemble-style methods such as FrugalGPT (Chen et al., 2023), RouterDC (Chen et al., 2024), and Avengers Pro (Zhang, 2025a;b) show that intelligently combining smaller models can approximate or even surpass larger frontier LMs. Decoder-based methods leverage a small language model to directly generate the routing decision. Causal LLM Routing, suggests that incorporating richer query–model interaction signals via generative modeling or cross-attention can yield more robust routing than static embeddings (Chen et al., 2024). In this work, we are inspired by these novel approaches to routing, and evaluate their performance in the local-cloud routing setup. In parallel, adaptive approaches such as PickLLM (Sikeridis et al., 2024), GraphRouter (Feng et al., 2025), and RadialRouter (Jin et al., 2025) explore reinforcement learning and graph-based methods to dynamically tailor routing policies based on context and resource constraints. These techniques highlight alternative designs beyond static routers, but often require additional feedback signals or structured representations. While we do not evaluate adaptive or RL-based routers in this work, we see them as a promising direction and leave their integration and benchmarking within local–cloud routing setups to future work.

**LLM Routing Benchmarks**    Recent work has explored benchmarks for LLM query routing, primarily targeting cost–quality tradeoffs across multiple models. RouterBench (Hu et al., 2024) provides a comprehensive suite of curated academic tasks ( 405K samples) to evaluate routing policies along cost–quality Pareto frontiers. RouteLLM (Ong et al., 2024) introduces a preference-trained routing framework evaluated on academic benchmarks like MMLU and MT-Bench, with a focus on achieving quality under token cost constraints, though it remains limited to token-level metrics. RouterEval (Huang et al., 2025) emphasizes model selection accuracy at scale, compiling over 200M performance records across 8.5K models and 12 benchmarks to study generalization, yet lacks coverage of real-world queries. In contrast, our proposed TRAFFICBENCH targets routing under naturalistic conditions, leveraging 1M real user queries from WILDCHAT and NATURALREASONING. It uniquely supports the exploration of local-cloud routing tradeoffs beyond just cost and quality, to metrics such as latency, energy, memory, throughput, and more — generated on local accelerators and enterprise-grade accelerators. Moreover, in contrast to existing benchmarks—which provide stale performance records limited to models released prior to July 2024—TRAFFICBENCH evaluates several state-of-the-art models, including Qwen3 (Team, 2025) and GPT-OSS (Agarwal et al., 2025), all released after May 2025. To support ongoing benchmarking, we release the TRAFFICBENCH Profiling Harness, a hardware-agnostic toolkit for generating fresh telemetry and evaluation records as new models become available.

**Local–Cloud Inference Systems**    Beyond model selection, recent work explores collaborative inference protocols that split generation between local and cloud models. Minions (Narayan et al., 2025) proposes a two-stage protocol where a small on-device LM handles lightweight processing and a frontier LM performs high-level reasoning, with an extended version introducing task decomposition and aggregation for improved quality. Such collaborative schemes offer large energy and cost savings but require careful protocol design to avoid performance loss. A parallel line of work centers on speculative decoding, where a small draft model generates candidate continuations that are verified or refined by a larger target LM (Miao et al., 2023; Xu et al., 2025). These approaches primarily target latency and throughput, particularly in constrained hardware settings, and typically assume that generation will ultimately invoke a large LM. Other hybrid protocols like SLED (Li et al., 2025) and HAT (Xie et al., 2025) introduce edge–cloud model partitioning with intermediate state exchange to balance device limitations with quality needs. While these systems explore fine-grained collaboration at the token or layer level, our work investigates the limitations

of a coarser-grained alternative: query-level routing across multiple small and large LMs, where we measure not only accuracy and cost, but also latency, memory, and energy across diverse hardware accelerators.

**Efficient AI**   We are inspired by work on "Green AI" which proposes treating energy as a first-class metric alongside accuracy and cost, with calls for standardized reporting and tooling for reproducible accounting of power use and emissions during training and inference (Schwartz et al., 2020; Strubell et al., 2019; Patterson et al., 2021; Henderson et al., 2020; Anthony et al., 2020; CodeCarbon Contributors, 2024; Oviedo et al., 2025). Complementary to our focus on local–cloud routing, cost and efficiency-driven model selection strategies such as FrugalML and FrugalGPT for API and model cascades, and CALM for token-wise early exit, dynamically allocate workloads to cheaper or smaller models while preserving quality (Chen et al., 2020; 2023; Schuster et al., 2022). On-device and edge studies demonstrate algorithm–hardware co-design for lower latency and energy consumption, exemplified by EdgeBERT's optimizations and PowerInfer's efficient LLM serving on commodity GPUs (Tambe et al., 2021; Song et al., 2024). Finally, hardware-aware benchmarking efforts such as "From Words to Watts" and MLPerf Power quantify inference energy across accelerators and standardize power measurement protocols (Samsi et al., 2023; Tschand et al., 2025).

## B   TRAFFICBENCH: DATASET AND PROFILING HARNESS

In this section, we provide additional details on the TRAFFICBENCH dataset.

### B.1   TRAFFICBENCH DATASET

Here, we provide additional details on the Anthropic Economic Index (Handa et al., 2025) categories used as labels in TRAFFICBENCH (see Table 5), the hardware platforms profiled, and the metrics recorded in TRAFFICBENCH.

| | |
|---|---|
| Life, physical, and social science | Computer and mathematical |
| Architecture and engineering | Education instruction and library |
| Installation, maintenance, and repair | Business and financial operations |
| Legal services | Transportation and material moving |
| Arts, design, sports, entertainment, and media | Production services |
| Farming, fishing, and forestry | Healthcare support |
| Food preparation and serving related | Healthcare practitioners and technical |
| Community and social service | Sales and related |
| Office and administrative support | General management |
| Protective service | Building grounds cleaning and maintenance |
| Construction and extraction | Personal care and service |

Table 5: **Anthropic Economic Index Categories (Handa et al., 2025)**. This taxonomy categorizes occupations into 22 standardized economic domains, adapted from U.S. Bureau of Labor Statistics frameworks. It is designed to support AI impact analysis by aligning labor categories with distinct task structures.

**Query Curation**   When sourcing queries from the WILDCHAT and NATURALREASONING datasets, we apply robust data cleaning and filtering to ensure the quality and consistency of the sampled queries. For NATURALREASONING, we filter out all queries that don't contain ground truth answers. For WILDCHAT, we eliminate non-English entries to maintain linguistic uniformity across the dataset. Queries that are malformed, nonsensical, or otherwise unintelligible (as determined by an LLM judge—i.e., GPT-4O-MINI) are discarded to prevent noise. Additionally, duplicate queries are removed to reduce redundancy and avoid overrepresentation of specific prompts. Finally, we filter out excessively long queries that exceed a 32,000-character limit.

**Dataset Statistics**   Table 6 reveals significant differences in how the two datasets in TRAFFICBENCH are distributed across domains. WILDCHAT is dominated by "Arts, design, sports, entertainment, and media" queries (47.1%), followed by "Computer and mathematical" (18.1%), while NATURALREASONING is primarily composed of "Life, physical, and social science" (36.0%) and "Computer and mathematical" (34.8%) queries. We use GPT-4O-MINI to bucket each query into its economic categorization using the prompt below.

```
You are a query categorizer. Your task is to categorize the following user
↪   query into one of the predefined categories based on the job/occupation
↪   domain it relates to most closely.

  Query: "{query}"

  Available Categories:
  − Office and administrative support
  − Transportation and material moving
  − Sales and related
  − Food preparation and serving related
  − General management
  − Business and financial operations
  − Healthcare practitioners and technical
  − Production services
  − Education instruction and library
  − Healthcare support
  − Construction and extraction
  − Installation, maintenance, and repair
  − Computer and mathematical
  − Building grounds cleaning and maintenance
  − Protective service
  − Personal care and service
  − Architecture and engineering
  − Community and social service
  − Arts, design, sports, entertainment, and media
  − Life, physical, and social science
  − Legal services
  − Farming, fishing, and forestry
  − None

  Instructions:
  1. Read the query carefully
  2. Determine which job/occupation category the query relates to most
  ↪   closely
  3. If the query doesn't clearly relate to any specific occupation category,
  ↪   use "None"
  4. Respond with ONLY the category name, exactly as listed above

  Category:
```

Solvability rates vary dramatically by domain and dataset type, where a query's solvability is defined as its ability to be answered correctly by any of the available local LMs (e.g. Qwen models or GPT OSS). WILDCHAT queries show consistently high solvability across most domains (generally ¿94%), with particularly strong performance in creative and social domains. In contrast, NATURALREASONING exhibits more variable solvability, with technical domains like "Architecture and engineering" showing only 41.5% solvability compared to 99.4% for the same domain in WILDCHAT. This disparity reflects the complexity difference between open-ended chat queries and analytical reasoning tasks, supporting our findings that chat queries are more amenable to local model routing than reasoning-intensive queries.

**Metrics** We detail all the metrics collected via our profiling harness in Table 7. For our correctness evaluations on WILDCHAT, we use an LLM-as-a-judge approach to evaluate model generated answers against a ground truth answer from GEMINI-2.5-PRO. For our correctness evaluations of NATURALREASONING, we use another LLM-judge prompt, but compare against ground truth answers provided in the original dataset. We provide both LLM-judge prompts below. The LLM used for each respective evaluation is GPT-4O.

**WILDCHAT LLM-judge Prompt**

```
You are an impartial judge evaluating the quality of two AI-assistant
↪   replies to the same user prompt.
```

| Domain | WC Count | WC % | WC Solv % | NR Count | NR % | NR Solv % |
|---|---|---|---|---|---|---|
| Computer and mathematical | 90,662 | 18.1 | 99.5 | 174,242 | 34.8 | 67.3 |
| Arts, design, sports, entertainment, and media | 235,658 | 47.1 | 98.7 | 2,648 | 0.5 | 52.9 |
| Life, physical, and social science | 28,079 | 5.6 | 98.8 | 180,065 | 36.0 | 60.5 |
| None | 49,014 | 9.8 | 97.3 | 79,752 | 16.0 | 65.6 |
| Education instruction and library | 23,196 | 4.6 | 97.2 | 13,864 | 2.8 | 80.4 |
| Architecture and engineering | 5,782 | 1.2 | 98.9 | 28,762 | 5.8 | 40.8 |
| Business and financial operations | 19,628 | 3.9 | 97.8 | 8,779 | 1.8 | 55.3 |
| Healthcare practitioners and technical | 8,905 | 1.8 | 98.1 | 1,851 | 0.4 | 66.3 |
| Office and administrative support | 6,959 | 1.4 | 91.6 | 25 | 0.0 | 48.0 |
| Legal services | 5,208 | 1.0 | 98.6 | 1,349 | 0.3 | 69.0 |
| Community and social service | 6,125 | 1.2 | 97.0 | 404 | 0.1 | 76.2 |
| Transportation and material moving | 1,890 | 0.4 | 95.1 | 3,914 | 0.8 | 53.0 |
| Sales and related | 4,689 | 0.9 | 97.8 | 218 | 0.0 | 67.4 |
| Food preparation and serving related | 3,308 | 0.7 | 98.3 | 507 | 0.1 | 61.9 |
| General management | 3,364 | 0.7 | 96.7 | 340 | 0.1 | 67.9 |
| Installation, maintenance, and repair | 954 | 0.2 | 97.1 | 2,037 | 0.4 | 56.2 |
| Farming, fishing, and forestry | 1,677 | 0.3 | 99.5 | 411 | 0.1 | 65.5 |
| Protective service | 1,315 | 0.3 | 97.9 | 237 | 0.0 | 56.5 |
| Construction and extraction | 991 | 0.2 | 97.2 | 147 | 0.0 | 60.5 |
| Healthcare support | 1,112 | 0.2 | 97.5 | 12 | 0.0 | 100.0 |
| Production services | 546 | 0.1 | 100.0 | 334 | 0.1 | 65.3 |
| Personal care and service | 648 | 0.1 | 92.9 | 19 | 0.0 | 0.0 |
| Building grounds cleaning and maintenance | 278 | 0.1 | 100.0 | 70 | 0.0 | 72.9 |
| **TOTAL** | **500K** | **100.0** | **98.4** | **500K** | **100.0** | **63.0** |

Table 6: **TRAFFICBENCH Domain Composition and LM Coverage ($\leq$20B Models)**. Comparison of domain distribution and model solvability rates across WILDCHAT (WC) and NATURALREASONING (NR) datasets. Solvability indicates the percentage of problems that can be solved correctly by at least one model with $\leq$20B parameters.

```
Step 1: Generate your own answer
Write the response *you* would give to the user. Keep it separate from later
↪  analysis.

Step 2: Decide the query type
Classify the user prompt as either
· **Subjective / open-ended** (creative writing, opinion, advice,
↪  brainstorming)
· **Objective / technical** (code, math, logical derivations with a single
↪  correct outcome)
If uncertain, default to "Subjective".

Step 3 { Score each assistant with the correct rubric

| Query type | Criteria |
|-----------|----------|
| Subjective / open-ended | 1. Correctness / factual soundness 2.
↪  Helpfulness 3. Relevance 4. Conciseness 5. Creativity & novelty |
| Objective / technical   | 1. Correctness only |

When using the multi-criteria rubric, note strengths and weaknesses for
↪  **each** dimension.
When using the single-criterion rubric, focus exclusively on factual /
↪  functional accuracy and ignore style or flair.

Step 4: Compare & justify
Explain which assistant is better and why, correcting any mistakes you find.
↪  Highlight missing but important details. **Be concise.**
```

```
Step 5:  Verdict
1. Assistant A is significantly better: [[A>>B]]
2. Assistant A is slightly better: [[A>B]]
3. Tie, Assistant A is equal: [[A=B]]
4. Assistant B is slightly better: [[B>A]]
5. Assistant B is significantly better: [[B>>A]]

Choose exactly one token from: `[[A>>B]]`, `[[A>B]]`, `[[A=B]]`, `[[B>A]]`,
↪  `[[B>>A]]`.

---

### Output format (strict)
Return **only** a JSON object that matches the provided schema:
\end{lstlisting}
```

**NATURALREASONING LLM-judge Prompt**

```
You are evaluating a response to a scientific/technical question against a
↪  reference answer.

Your task is to determine if the response is factually correct and complete
↪  compared to the reference.

Consider:
1. Scientific accuracy of facts and concepts
2. Mathematical correctness (if applicable)
3. Completeness of the answer
4. Technical precision

Question: {question}

Response: {response}

Reference Answer: {reference}

Return ONLY 'True' if the response is correct and complete, 'False'
↪  otherwise.
```

**Hardware Backends**    Details regarding profiled hardware can be found in Table 8.

**Data Generation Procedure**    We generate model outputs using consistent decoding settings across all tasks: temperature = 0.6, top-p = 0.95, top-k = 20, min-p = 0.0, and a 4096-token output limit. For NATURALREASONING queries, we enable deliberative prompting (use thinking = True); for WILDCHAT, we disable it. For QWEN models, we apply a repetition penalty of 1.1 and length penalty of 1.0.

**Telemetry Collection**    We collected telemetry by instrumenting host-level samplers that interface directly with vendor-supported system APIs on each platform. Data were obtained from NVML on NVIDIA-equipped hosts, from the powermetrics facility on macOS, and from ROCm SMI on AMD-equipped hosts. Each sampler queried the respective system interface to obtain GPU- and system-level measurements and produced synchronized records suitable for downstream quantitative analysis.

On NVIDIA systems, we interface directly with NVML and enumerate all visible GPUs. For each device, we query instantaneous power as reported by the driver, read cumulative energy from the on-device counter, obtain GPU temperature from the hardware sensor, and retrieve memory usage from the device's memory interface. Units are normalized (e.g., milliwatts and millijoules mapped to watts and joules; bytes to megabytes). In multi-GPU hosts, power and memory are summed across devices and temperature is averaged to yield a single aggregate view. Each record also includes host memory usage from OS counters, a nanosecond timestamp, and device identity and backend provenance.

| Metric | Description |
|---|---|
| `flops_per_request` | FLOPs per query. |
| `macs_per_request` | MACs per query; proxy for compute. |
| `per_query_joules` | Energy per query (J). |
| `total_joules` | Total energy across queries. |
| `per_token_ms` | Latency per token (ms). |
| `throughput_tokens_per_sec` | Token output rate (toks/s). |
| `time_to_first_token_seconds` | Time to first token (s). |
| `total_query_seconds` | Total time per query (s). |
| `cpu_mb.avg / max / median / min` | CPU memory usage (MB). |
| `gpu_mb.avg / max / median / min` | GPU memory usage (MB). |
| `initialization_duration_seconds` | Model load time (s). |
| `batch_size` | Query batch size. |
| `gpu_memory_utilization` | GPU memory use (0–1). |
| `max_model_len` | Max token length allowed. |
| `max_num_batched_tokens` | Max batch token count. |
| `max_output_tokens` | Max output tokens. |
| `num_workers` | Number of threads. |
| `temperature` | Sampling temperature. |
| `top_k` | Top-k cutoff. |
| `top_p` | Top-p (nucleus) threshold. |
| `warmup_steps` | Warm-up steps. |
| `per_query_watts.avg / max / median / min` | GPU power draw per query (W). |
| `total_watts.avg / max / median / min` | Session GPU power draw (W). |
| `cpu_count` | CPU core count. |
| `cpu_brand` | CPU model. |
| `host_name` | Machine hostname. |
| `os_name / os_version / kernel_version` | OS and kernel info. |
| `temperature.avg / max / median / min` | Device temperature (°C). |
| `input` | Input tokens per query. |
| `output` | Output tokens per query. |

Table 7: **TRAFFICBENCH metrics**. Summary of compute, latency, memory, and energy profiling metrics.

| Hardware | Memory | Bandwidth | Power |
|---|---|---|---|
| NVIDIA A100 40 GB SXM4 (Ampere) | 40 GB HBM2 | 1,555 GB/s | 400 W TDP |
| NVIDIA H200 SXM (Hopper) | 141 GB HBM3e | 4.8 TB/s | Up to 700 W TDP |
| NVIDIA B200 (Blackwell) | 192 GB HBM3e | 8 TB/s | 1000 W TDP |
| NVIDIA GH200 (Grace Hopper) | 144 GB HBM3e + 624 GB LPDDR5X | 4.8 TB/s (GPU) | 1000 W TDP |
| AMD Instinct MI300X (CDNA 3, OAM) | 192 GB HBM3 | 5.3 TB/s (peak) | 750 W TBP |
| Apple Mac Studio (M4 Max) | 128 GB unified | 546 GB/s | 480 W (continuous) |
| NVIDIA Quadro RTX 6000 (Turing) | 24 GB GDDR6 | 672 GB/s | 295 W TDP |
| NVIDIA RTX 6000 Ada Generation | 48 GB GDDR6 | 960 GB/s | 300 W TDP |
| Apple iPhone 16 Pro (A18 Pro, integrated NPU) | 8 GB LPDDR5X unified | ∼60 GB/s | ∼12 W (SoC peak, 35 TOPS NPU) |
| SambaNova SN40L RDU | 64 GB HBM2E | 1.6 TB/s | 500 W TDP |

Table 8: **Accelerator Details**. Memory, bandwidth, and power specifications of evaluated accelerators and systems.

On macOS, we execute `powermetrics` with elevated privileges and ingest its continuous `plist` stream. Each `plist` frame is parsed to extract the GPU power value exposed by the system (Apple Silicon: `processor_power.actual`; Intel: `processor.combined_power`), which is normalized to watts. Energy (joules) is obtained by numerically integrating the power signal over successive frames using the measured inter-frame wall-clock interval. In parallel, system memory usage is sampled from OS counters. Every observation is timestamped and annotated with Apple device identity and an explicit `powermetrics` backend tag.

On AMD systems, we use ROCm SMI to query current GPU power (watts), read temperature from junction or edge sensors (°C), and obtain VRAM usage from the device memory interface (bytes to megabytes). Energy (joules) is computed by integrating the power signal over time using consecutive

| Size Threshold ($\leq$) | Cost Savings | | Compute Savings | | Energy Savings | |
|---|---|---|---|---|---|---|
| | Qwen + GPT-OSS | Qwen | Qwen + GPT-OSS | Qwen | Qwen + GPT-OSS | Qwen |
| 4B | 65.2% | 65.2% | 65.1% | 65.1% | 63.5% | 63.5% |
| 8B | 80.8% | 80.8% | 83.1% | 83.1% | 79.6% | 79.6% |
| 14B | 89.0% | 89.0% | 93.0% | 93.0% | 87.0% | 87.0% |
| 20B | 90.5% | — | 97.4% | — | 89.4% | — |
| 32B | 91.3% | 91.9% | 97.4% | 92.8% | 90.4% | 90.5% |
| 120B | 91.1% | — | 97.7% | — | 90.7% | — |

Table 9: **Cost, Compute, and Energy Savings from Local-Cloud Routing on WildChat**: Savings across different resources while maintaining task accuracy of SOTA open-source cloud model (i.e. `Qwen3 235B-A22B`).

| Size Threshold ($\leq$) | Cost Savings | | Compute Savings | | Energy Savings | |
|---|---|---|---|---|---|---|
| | Qwen + GPT-OSS | Qwen | Qwen + GPT-OSS | Qwen | Qwen + GPT-OSS | Qwen |
| 4B | 52.9% | 52.9% | 54.5% | 54.5% | 46.3% | 46.3% |
| 8B | 60.5% | 60.5% | 62.5% | 62.5% | 54.0% | 54.0% |
| 14B | 68.7% | 68.7% | 70.1% | 70.1% | 62.5% | 62.5% |
| 20B | 73.3% | — | 72.2% | — | 67.8% | — |
| 32B | 76.9% | 75.9% | 75.1% | 75.1% | 72.4% | 71.6% |
| 120B | 86.7% | — | 86.0% | 86.0% | 85.9% | — |

Table 10: **Cost, Compute, and Energy Savings from Local-Cloud Routing on NaturalReasoning**: Savings across different resources while maintaining task accuracy of SOTA open-source cloud model (i.e. `Qwen3 235B-A22B`).

sampling intervals. System memory usage is read from OS counters. In multi-GPU machines, the primary device under observation is explicitly selected (GPU index 0 in our setup), and all records carry precise timestamps together with device identity and backend metadata.

## C ROUTING METHODOLOGIES

### C.1 EMBEDDING-BASED ROUTING

**Hyperparameter Search** We identify the optimal hyperparameters for each of the embedding-based architectures by performing a comprehensive grid search across learning rates ($\{1e^{-6}, 5e^{-6}, 1e^{-5}, 5e^{-5}\}$), batch sizes ($\{16, 32, 64, 128\}$), and for the k-NN approach, neighborhood sizes ($k \in \{1, 5, 10, 50, 100\}$). For the finetuned embedding classifier, we additionally sweep over MLP hidden dimensions ($\{128, 256, 512\}$) and dropout rates ($\{0.1, 0.2, 0.3\}$). We use full supervised finetuning for all embedding models explored: ANSWERDOTAI/MODERNBERT-BASE (Warner et al., 2024), QWEN/QWEN3-EMBEDDING-0.6B, QWEN/QWEN3-EMBEDDING-4B, and QWEN/QWEN3-EMBEDDING-8B (Zhang et al., 2025). All models are trained for 10 epochs with early stopping based on validation F1 score (patience of 3 epochs). We employ linear warmup for the first 10% of training steps followed by cosine decay. The optimal configuration achieved uses a learning rate of $5e^{-6}$, batch size of 64, and for k-NN, $k = 100$ neighbors (Table 11).

### C.2 DECODER-BASED ROUTING

**Hyperparameter Search** We identify the optimal hyperparameters for the decoder-based architectures through systematic evaluation of learning rates ($\{1e^{-6}, 5e^{-6}, 1e^{-5}\}$), batch sizes ($\{8, 16, 32, 64\}$), confidence thresholds ($\tau \in \{0.3, 0.5, 0.7, 0.9\}$), and decoder backbone sizes ($\{0.6B, 1.5B, 3B, 8B\}$ parameters). For both multi-class and binary classification formulations, we use full supervised finetuning for all decoder models tested: QWEN/QWEN3-0.6B, QWEN/QWEN3-4B, and QWEN/QWEN3-8B (Team, 2025). Models are trained for 5 epochs with gradient accumulation steps adjusted to maintain an effective batch size of 64. We implement linear warmup for 10% of total steps followed by cosine decay. The optimal configuration uses a learning rate of $5e^{-6}$, effective batch size of 64, confidence threshold $\tau = 0.5$, and the 8B parameter decoder backbone for all main experiments.

| k | WildChat | NaturalReasoning |
|---|----------|------------------|
| 1 | 0.758 | 0.681 |
| 5 | 0.781 | 0.714 |
| 10 | 0.805 | 0.748 |
| 50 | 0.841 | 0.783 |
| 100 | **0.875** | **0.827** |
| 500 | 0.821 | 0.804 |
| 1000 | 0.834 | 0.747 |

Table 11: **Performance of k-NN routing on Query Embeddings**: Values represent macro F1 scores.

## D  TRAFFICBENCH EXPERIMENTS

### D.1  REAL-WORLD EFFICIENCY AND DEPLOYMENT IMPACT

We compute cost per query using pricing available on OpenRouterAI (OpenRouter, 2025). Table 12 lists the token pricing used.

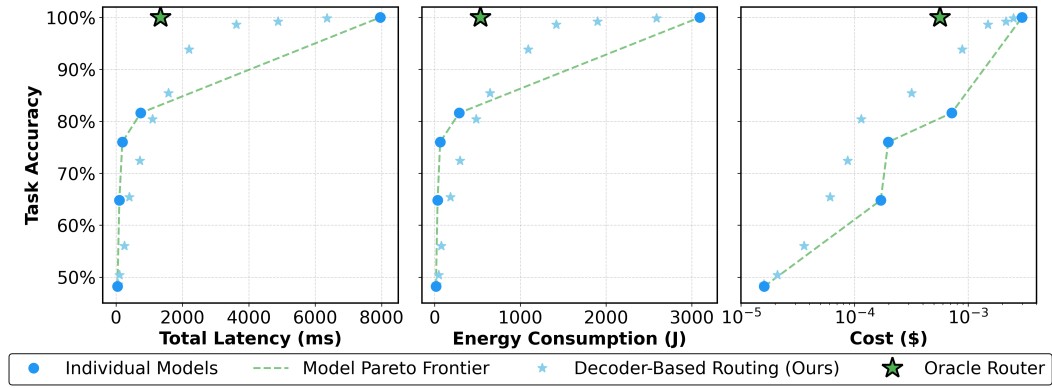

Figure 6: **Accuracy Tradeoffs for Model Routing**. Decoder-based routing achieves Pareto-optimal tradeoffs across latency, cost and energy. Individual points show fixed deployments of small models deployed locally (4B-20B) and frontier models in the cloud (235B), while oracle routing represents the theoretical optimum of always selecting the smallest capable configuration.

| Model | Input Cost (USD / 1M tokens) | Output Cost (USD / 1M tokens) |
|-------|------------------------------|-------------------------------|
| Qwen3-4B | 0.000 | 0.000 |
| Qwen3-8B | 0.035 | 0.138 |
| Qwen3-14B | 0.060 | 0.124 |
| Qwen3-32B | 0.100 | 0.450 |
| Qwen3-235B | 0.220 | 0.880 |

Table 12: **Model pricing from OpenRouterAI.** Costs are in USD per 1M tokens for input and output, as of August 2025.

### D.2  HOW DOES MODEL PRECISION AFFECT PERFORMANCE AND EFFICIENCY?

Model quantization—reducing numerical precision from FP16 to FP8 or FP4—decreases memory requirements and energy consumption during inference while introducing approximation error that may degrade model accuracy. To quantify this tradeoff, we evaluate eight open-source models from the QWEN3 and GEMMA families across three precision levels: FP16 (full precision), FP8 (8-bit floating point), and FP4 (4-bit floating point). For each model-precision pair, we measure accuracy on three reasoning-focused datasets: NATURALREASONING ($N=10,000$), SuperGPQA ($N=10,000$), and MMLU Pro ($N=10,000$).

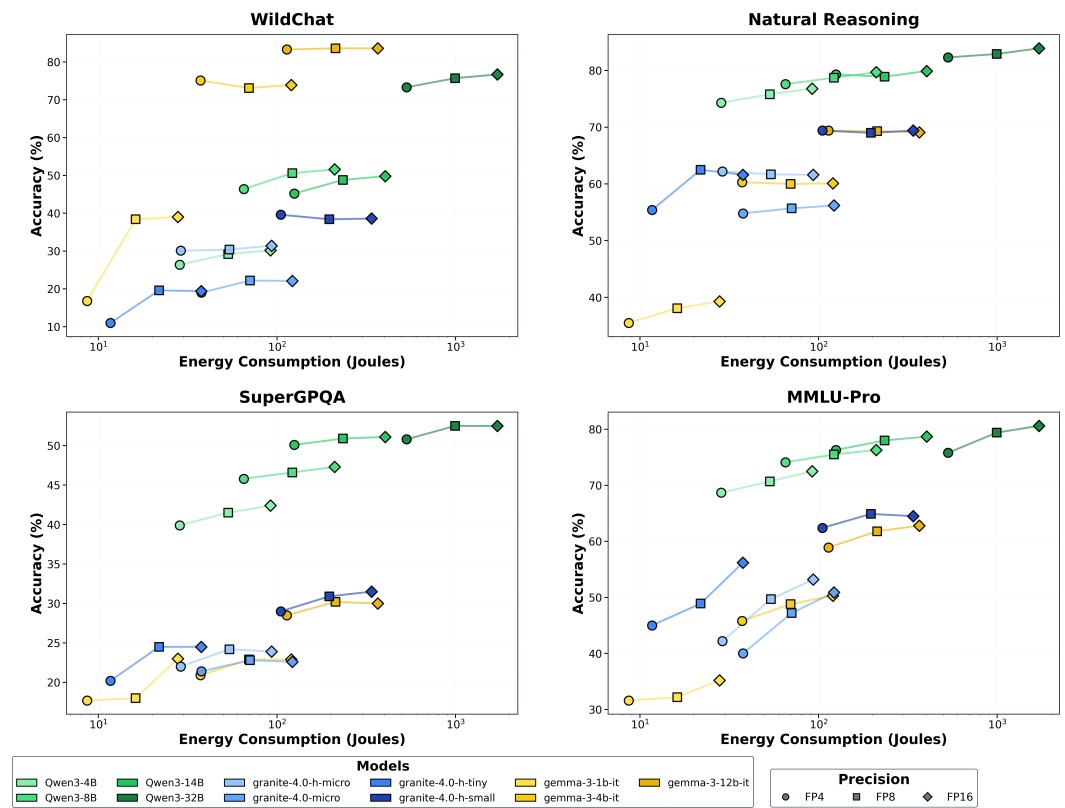

Figure 7: **Minimal Accuracy Degradation Shifting from FP16 to FP4 for Open-Source Local Models**: Evaluation across three reasoning datasets ($N = 10,000$ each) shows $2-3\%$ accuracy loss per precision step, demonstrating that $FP8/FP4$ quantization enables efficient deployment with acceptable performance tradeoffs.

Figure 7 shows that quantization from FP16 to FP4 yields energy reductions of $3-3.5\times$ with accuracy degradation of approximately 2.5 percentage points per precision step across all models and datasets. For example, on SuperGPQA, QWEN3-14B achieves 54.5% (FP16), 52.0% (FP8), and 49.0% (FP4)—a total degradation of 5.5 percentage points despite a $3.23\times$ reduction in energy consumption. Larger models maintain their relative performance advantage even at lower precision: QWEN3-14B at FP4 (49.0% accuracy) outperforms QWEN3-4B at FP16 (48.5% accuracy) on SuperGPQA, indicating that model scale matters more than precision for reasoning tasks. These results demonstrate that FP8 and FP4 quantization enable practical deployment of local models with predictable performance tradeoffs, allowing system designers to select precision levels based on application-specific requirements while capturing most of the energy savings identified in Section 5.4.

### D.3 RELIABILITY OF LLM JUDGES

Table 13: Individual Judge Agreement with Majority Vote

| Judge | Pearson | Cohen's $\kappa$ | Agreement |
|-------|---------|------------------|-----------|
| Claude Sonnet 4.5 | 0.744 | 0.736 | 89.7% |
| Grok-4 | 0.797 | 0.796 | 91.8% |
| GPT-4o | 0.692 | 0.677 | 85.4% |

We evaluated the reliability of three LLM judges (GPT-4O, CLAUDE SONNET 4.5, and GROK-4) on 1000 random samples from the WILDCHAT dataset. For each answer generated by the 8 core models evaluated (QWEN3-14B, QWEN3-235B-A22B, QWEN3-30B-A3B, QWEN3-32B, QWEN3-4B, QWEN3-8B,

GPT-OSS-120B, and GPT-OSS-20B), each judge independently classified model responses as correct (equal to or better than reference) or incorrect based on the prompt in App B.1.

We conduct an ablation study measuring its agreement rate of each individual judge with a stronger judge baseline: the majority vote across all three LLM judges. We find that individual judges showed strong alignment with majority vote decisions (Table 13): Grok-4 ($\kappa = 0.796$, 91.8% agreement), Claude Sonnet 4.5 ($\kappa = 0.736$, 89.7% agreement) and GPT-4o ($\kappa = 0.677$, 85.4% agreement). GPT-4o's substantial $\kappa$ value ($> 0.6$) indicates reliable performance that justifies its widespread use in the community.

### D.4 How does performance change when using different reference responses?

To investigate the impact of reference model selection on local LM coverage, we conduct a systematic ablation study using GPT-4O as a judge to compare model responses against three different reference models: CLAUDE SONNET 4.5, GEMINI 2.5 PRO, and GPT-5 (2025-08-07). Using the WILDCHAT dataset, we evaluate four local models (Qwen3-4B, Qwen3-8B, Qwen3-14B, and GPT-OSS-20B). For each problem, the judge performs pairwise comparisons between each evaluated model's response as either correct (outperforms the reference model) or in-correct (is worse than reference response) using the LLM-as-a-judge evaluation criteria found in App. B.1. We compute local coverage across all the models evaluated, and find that accuracy when using GEMINI 2.5 PRO is 2.4 pp higher than GPT-5 as a reference model and 5.1 percentage points lower than CLAUDE SONNET 4.5 as a model. This indicates that while CLAUDE SONNET 4.5 yields somewhat higher measured local coverage, the choice of reference model has a relatively modest impact, with all three reference models producing local accuracies within a 7.5 percentage point range. These results suggest that local model coverage estimates are robust to the choice of frontier model responses used as a reference.

### D.5 What are the intelligence efficiencies of model / hardware pairs on TrafficBench? Which are the most power efficient and energy efficient?

To understand the hardware-specific efficiency characteristics of local versus cloud deployment, we evaluate intelligence efficiency—defined as task accuracy per unit of energy—across six accelerator platforms spanning mobile, local workstation, and cloud infrastructure. Table 14 reveals that cloud accelerators achieve 1.4–1.8$\times$ higher intelligence per watt than local workstation accelerators for identical models, with the NVIDIA B200 reaching $2.75 \times 10^{-3}$ accuracy/watt for Qwen3-32B compared to $1.97 \times 10^{-3}$ on the Apple M4 Max. Specialized cloud architectures like the SambaNova SN40L demonstrate the largest efficiency gains at $3.51 \times 10^{-3}$ accuracy/watt. However, mobile hardware exhibits exceptional power efficiency when accounting for quantization: the iPhone 16 Pro achieves $19.8$–$23.7 \times 10^{-3}$ accuracy/watt across Qwen3 models at FP4/FP8 precision—an order of magnitude higher than workstation GPUs—though this comes at the cost of reduced numerical precision and memory capacity constraints that limit model selection. Workstation GPUs (RTX 6000 Ada, MI300X) show competitive per-watt efficiency ($1.5$–$2.1 \times 10^{-3}$) but suffer from 1.6–2.0$\times$ lower efficiency than cloud accelerators due to architectural differences in memory bandwidth and compute optimization.

When evaluating intelligence per joule—which incorporates both power efficiency and latency—the efficiency gaps widen substantially to 1.6–7.4$\times$ in favor of cloud infrastructure. The SambaNova SN40L achieves $22.7$–$31.2 \times 10^{-5}$ accuracy/joule, outperforming the M4 Max baseline by 6.5–7.4$\times$ for large models, while the NVIDIA B200 provides 1.6–2.3$\times$ gains. This per-joule disadvantage for local accelerators stems from higher inference latency that increases total energy consumption despite comparable instantaneous power draw. Critically, however, these raw efficiency metrics do not determine deployment strategy: despite cloud accelerators' superior per-query efficiency, our routing analysis (Section 5.4) demonstrates that local deployment combined with intelligent routing enables 88.7% of queries to avoid cloud infrastructure entirely, yielding 60–80% resource savings system-wide. This result highlights that *workload-aware routing dominates hardware efficiency*—the ability to correctly identify and route the majority of queries to adequate local models produces larger aggregate savings than the marginal per-query efficiency gains from always using optimized cloud hardware.

### D.6 How does $\tau$ affect routing performance?

The confidence threshold $\tau$ plays a critical role in balancing routing quality, task accuracy, and efficiency gains. We conduct a systematic sweep of $\tau \in \{0.1, 0.2, ..., 0.9\}$ to quantify its impact on routing decisions

Table 14: **Intelligence Efficiency Across Hardware Platforms.** Intelligence per watt (accuracy/W $\times 10^{-3}$, top) shows cloud accelerators achieve $1.4-1.8\times$ higher efficiency than local accelerators for identical models. Intelligence per joule (accuracy/J $\times 10^{-5}$, bottom) demonstrates even larger efficiency gaps $(1.6-7.4\times)$ when accounting for latency, with specialized architectures (SN40L) achieving the largest gains. Workstation GPUs (RTX 6000 Ada, MI300X) show competitive per-watt efficiency but poor per-joule performance due to higher latency. Despite cloud efficiency advantages, local deployment enables 88.7% of queries to avoid cloud infrastructure entirely, yielding $60-80\%$ resource savings through intelligent routing. For iPhone 16 Pro: *FP16 precision. †FP8 precision. ‡FP4 precision.

| Model | iPhone 16 Pro (Local) | M4 Max (Local) | RTX 6000 (Local) | MI300X (Local) | B200 (Cloud) | SN40L (Cloud) |
|---|---|---|---|---|---|---|
| *Intelligence per Watt ($\times 10^{-3}$)* | | | | | | |
| Qwen3-4B | 19.8* | 1.40 | 1.52 | 1.48 | 1.95 | — |
| Qwen3-8B | 22.8† | 1.63 | 1.75 | 1.70 | 2.27 | — |
| Qwen3-14B | 23.7‡ | 1.69 | 1.82 | 1.76 | 2.35 | — |
| Qwen3-32B | — | 1.97 | 2.09 | 2.03 | 2.75 | 3.51 |
| **Efficiency Gap** | 13× | 1.0× | 1.1× | 1.05× | 1.4× | 1.8× |
| *Intelligence per Joule ($\times 10^{-5}$)* | | | | | | |
| Qwen3-8B | 2.85† | 3.80 | 2.15 | 1.98 | 8.71 | — |
| Qwen3-32B | — | 3.51 | 1.89 | 1.73 | 5.91 | 22.7 |
| GPT-OSS-20B | — | 4.38 | 2.52 | 2.31 | 7.34 | — |
| GPT-OSS-120B | — | 4.23 | 2.41 | 2.19 | 6.78 | 31.2 |
| **Efficiency Gap** | 0.75× | 1.0× | 0.5−0.6× | 0.45−0.55× | 1.6−2.3× | 6.5−7.4× |

and downstream metrics. Table 15 reveals that routing F1 peaks at $\tau = 0.5$, achieving 0.851 with 85.4% task accuracy and 77.1% energy savings. Lower thresholds ($\tau < 0.5$) improve routing F1 initially but sacrifice task accuracy—at $\tau = 0.3$, routing F1 reaches 0.718 but task accuracy drops to 56.0%. Conversely, higher thresholds ($\tau > 0.5$) maximize task accuracy at the expense of routing quality: $\tau = 0.7$ achieves 94.5% task accuracy but routing F1 degrades to 0.763 as the router becomes overly conservative, routing 74.8% of queries to the cloud. Table 16 demonstrates how $\tau$ controls the allocation of queries across model tiers. At aggressive thresholds ($\tau = 0.1$), 94.2% of queries route to the smallest 4B model, yielding a weighted average of only 4.1B parameters but achieving just 48.2% task accuracy. As $\tau$ increases to 0.5, the router balances 42.1% on 4B, 24.3% on 8B, 14.6% on 14B, 9.2% on 20B, and 9.8% on cloud (235B), with a weighted average of 18.7B.

Table 17 characterizes three operationally distinct regimes. The **aggressive regime** ($\tau = 0.3$) maximizes efficiency with 97.5% energy savings and 98.8% cost reduction by routing 97.4% of queries locally, but sacrifices task accuracy (56.0%)—suitable for low-stakes, efficiency-critical applications. The **balanced regime** ($\tau = 0.5$) achieves production-grade performance with 85.4% task accuracy, 77.1% energy savings, 60.2% cost reduction, and 90.2% local routing—optimal for most real-world deployments. The **conservative regime** ($\tau = 0.7$) prioritizes accuracy (94.5%) over efficiency, yielding only 16.3% energy and 15.8% cost savings—appropriate for high-stakes tasks requiring maximum correctness. Our experiments identify $\tau = 0.5$ as the optimal threshold, capturing 78.1% of oracle routing efficiency while maintaining practical latency

### D.7 HOW DOES ROUTING COMPARE ACROSS DOMAINS, DIFFICULTIES, AND QUERY LENGTHS?

Routing performance varies systematically across query characteristics. Table 18 reveals that domain-specific complexity drives distinct failure modes: creative and social domains (Arts & Media, Community Service) achieve 62% correct routing with only 8–10% undersizing, while technical domains (Architecture, Life/Physical Sciences) suffer 50–58% undersizing with just 31–36% correct routing. Query difficulty exhibits a monotonic relationship with routing accuracy—easy queries (solvable by 4 models) achieve 60% correct routing with 31% wasteful oversizing, while unsolvable queries yield only 32% correct routing with 59% undersizing, indicating the router struggles to recognize inherently difficult problems. Query length demonstrates an inverse correlation with routing quality: short queries (<50 tokens) achieve 60% correct routing but suffer 32% oversizing, while long queries (>1000 tokens) drop to 34% correct routing with 58% undersizing, suggesting the router underestimates complexity when contextual information spans many tokens.

Table 19 illustrates representative failure modes and success cases. **Under-routing failures** occur when the router underestimates task complexity: Example 1 shows an 8B model failing on multi-step thermodynamics requiring equation coupling (oracle: 32B), Example 3 demonstrates 4B inadequacy for

Table 15: **Routing F1 peaks at $\tau = 0.5$ with 85.4% task accuracy and 77.1% energy savings.** Analysis of confidence threshold $\tau$ impact on routing performance across accuracy, efficiency metrics, and savings compared to cloud-only deployment (Qwen3-235B) under real-world traffic distribution (Wang et al., 2025). Routing F1 measures routing decision quality; Task Acc measures end-task accuracy.

| $\tau$ | Accuracy | | Absolute Metrics | | | Savings vs Cloud | | |
|---|---|---|---|---|---|---|---|---|
| | Routing F1 | Task Acc (%) | Avg Size (B) | Energy (kJ) | Cost ($) | Energy (%) | Cost (%) | Compute (%) |
| 0.1 | 0.542 | 48.2 | 4.1 | 0.015 | 0.000016 | 99.5 | 99.5 | 99.3 |
| 0.2 | 0.631 | 48.6 | 5.2 | 0.025 | 0.000016 | 99.2 | 99.5 | 98.9 |
| 0.3 | 0.718 | 56.0 | 7.8 | 0.077 | 0.000036 | 97.5 | 98.8 | 96.2 |
| 0.4 | 0.789 | 72.4 | 12.3 | 0.298 | 0.000087 | 90.4 | 97.1 | 88.7 |
| **0.5** | **0.851** | **85.4** | **18.7** | **0.708** | **0.001188** | **77.1** | **60.2** | **67.1** |
| 0.6 | 0.824 | 90.6 | 45.2 | 1.420 | 0.001495 | 54.1 | 49.9 | 51.6 |
| 0.7 | 0.763 | 94.5 | 98.4 | 2.590 | 0.002513 | 16.3 | 15.8 | 18.1 |
| 0.8 | 0.691 | 100.0 | 187.5 | 3.094 | 0.002985 | 0.1 | 0.0 | 0.2 |
| 0.9 | 0.587 | 100.0 | 229.8 | 3.096 | 0.002986 | 0.0 | 0.0 | 0.0 |
| *Reference baselines:* | | | | | | | | |
| Cloud-only | – | 100.0 | 235.0 | 3.096 | 0.002986 | 0.0 | 0.0 | 0.0 |
| Oracle | 1.000 | 100.0 | 15.2 | 0.535 | 0.000565 | 82.7 | 81.1 | 78.1 |

Table 16: **At optimal threshold ($\tau = 0.5$), 90.2% of queries route to local models ($\leq 20B$).** Query distribution across models for different confidence thresholds, showing percentage of queries routed to each model size at varying $\tau$ values. Lower thresholds favor smaller models; higher thresholds route more conservatively to larger models. Weighted average model size increases monotonically with $\tau$.

| $\tau$ | 4B | 8B | 14B | 20B | Cloud (235B) | Weighted Avg (B) |
|---|---|---|---|---|---|---|
| 0.1 | 94.2% | 3.1% | 1.5% | 0.8% | 0.4% | 4.1 |
| 0.2 | 88.5% | 5.8% | 2.9% | 1.6% | 1.2% | 5.2 |
| 0.3 | 76.3% | 11.2% | 6.4% | 3.5% | 2.6% | 7.8 |
| 0.4 | 58.7% | 18.5% | 10.8% | 6.2% | 5.8% | 12.3 |
| **0.5** | **42.1%** | **24.3%** | **14.6%** | **9.2%** | **9.8%** | **18.7** |
| 0.6 | 18.5% | 15.8% | 12.4% | 18.6% | 34.7% | 45.2 |
| 0.7 | 5.2% | 6.8% | 4.9% | 8.3% | 74.8% | 98.4 |
| 0.8 | 0.8% | 1.2% | 0.9% | 1.5% | 95.6% | 187.5 |
| 0.9 | 0.2% | 0.3% | 0.2% | 0.4% | 98.9% | 229.8 |

*Note: Percentages show fraction of 100K test queries routed to each model.*

nuanced creative writing with atmospheric detail (oracle: 32B), and Example 5 reveals 8B insufficiency for multi-factor constitutional analysis (oracle: 32B). **Over-routing waste** occurs when the router overestimates difficulty: Example 2 routes simple one-step arithmetic to 32B when 4B suffices, triggered by the phrase "show your work." **Context-dependent failures** arise when queries reference missing conversation state (Example 4), making single-turn routing fundamentally insufficient regardless of model selection. **Success cases** (Example 6) demonstrate that the router correctly identifies standard algorithm implementations as 8B-appropriate, avoiding both under-routing (correctness risk) and over-routing (efficiency waste). These patterns suggest that improved routing requires better calibration for long-form queries, technical complexity detection, and multi-turn context awareness.

### D.8 WHAT IS THE LATENCY AND ENERGY DEMANDS OF THE ROUTER?

The routing decision overhead represents a modest fraction of end-to-end inference costs. Table 20 quantifies the computational cost of our decoder-based router (Qwen3-8B) deployed on an NVIDIA H200 across different workload types. Router latency averages 39.1ms per batch, constituting 14.9% of total latency overhead, while router energy consumption averages 1,119J per batch, representing 8.4% of total energy overhead. Overhead varies by workload complexity: chat queries (WildChat) incur lower routing costs with 12.9% latency and 7.1% energy overhead, while reasoning queries (NaturalReasoning)

Table 17: **Three deployment regimes optimize for different objectives: efficiency ($\tau$=0.3), balance ($\tau$ = 0.5), or accuracy ($\tau$ = 0.7).** Comparison of routing behavior and efficiency tradeoffs at three representative threshold values under real-world traffic distribution (Wang et al., 2025).

| **Metric** | $\tau$=0.3 (Aggressive) | $\tau$=0.5 (Balanced) | $\tau$=0.7 (Conservative) |
|---|---|---|---|
| Task Accuracy | 56.0% | 85.4% | 94.5% |
| Local Query % | 97.4% | 90.2% | 25.2% |
| Energy Savings | 97.5% | 77.1% | 16.3% |
| Cost Savings | 98.8% | 60.2% | 15.8% |
| Avg Latency (ms) | 255 | 1,583 | 6,360 |
| Routing F1 | 0.718 | 0.851 | 0.763 |
| **Use Case** | Max efficiency Low stakes | Production Balanced | Max accuracy High stakes |

Table 18: **Failure Modes across TRAFFICBENCH Router by Domain, Query Difficulty, and Query Length**

| | **By Domain** | | |
|---|---|---|---|
| **Domain** | **Oversized Model %** | **Correct Model %** | **Undersized Model %** |
| Office & Admin | 26 | 56 | 18 |
| Transportation | 13 | 38 | 49 |
| Sales & Related | 27 | 61 | 12 |
| Food Prep | 25 | 58 | 17 |
| General Mgmt | 27 | 59 | 14 |
| Business & Finance | 18 | 45 | 37 |
| Healthcare Prac. | 25 | 53 | 22 |
| Production Svc | 18 | 44 | 38 |
| Education | 19 | 47 | 34 |
| Healthcare Support | 31 | 65 | 4 |
| Construction | 21 | 52 | 27 |
| Install & Repair | 15 | 40 | 45 |
| Computer & Math | 19 | 45 | 36 |
| Building Grounds | 26 | 60 | 14 |
| Protective Svc | 21 | 49 | 30 |
| Personal Care | 22 | 52 | 26 |
| Architecture | 11 | 31 | 58 |
| Community Svc | 28 | 62 | 10 |
| Arts & Media | 30 | 62 | 8 |
| Life/Physical Sci | 14 | 36 | 50 |
| Legal Services | 24 | 53 | 23 |
| Farming/Forestry | 20 | 50 | 30 |
| *Average* | 22 | 51 | 27 |

| | **By Difficulty** | | |
|---|---|---|---|
| **# Models** | **Oversized Model %** | **Correct Model %** | **Undersized Model %** |
| 0 (Unsolvable) | 9 | 32 | 59 |
| 1 (Very Hard) | 16 | 38 | 46 |
| 2 (Hard) | 21 | 50 | 29 |
| 3 (Medium) | 24 | 57 | 19 |
| 4 (Easy) | 31 | 60 | 9 |
| *Average* | 20 | 47 | 33 |

| | **By Query Length** | | |
|---|---|---|---|
| **Tokens** | **Oversized Model %** | **Correct Model %** | **Undersized Model %** |
| < 50 | 32 | 60 | 8 |
| 50–100 | 28 | 58 | 14 |
| 100–200 | 22 | 52 | 26 |
| 200–500 | 18 | 46 | 36 |
| 500–1000 | 12 | 40 | 48 |
| > 1000 | 8 | 34 | 58 |
| *Average* | 20 | 48 | 32 |

require more router computation at 17.0% latency and 9.8% energy overhead, reflecting the router's need to process more complex query features when making routing decisions for technical content.

These overhead costs are substantially outweighed by the efficiency gains from successful routing. Comparing router overhead (8.4% energy) against the 77.1% energy savings achieved through local-cloud routing (Table 15), the net benefit remains strongly positive—routing yields a 9.2× return on invested router energy. Furthermore, the router operates at batch granularity, amortizing its fixed inference cost across multiple queries, making the per-query overhead negligible in production deployments with typical batch sizes of 32–64. These results demonstrate that decoder-based routing with an 8B parameter model achieves practical efficiency tradeoffs, with routing overhead remaining below 15% while enabling order-of-magnitude improvements in system-wide energy and cost efficiency.

Table 19: **Qualitative Analysis of Routing Decisions**: Success Cases and Failure Modes

| **Example 1: Technical Under-routing** (×) |
|---|
| **Query:** *"Given a cylindrical vessel with a friction-less piston (mass 9kg, area 0.09$m^2$), initial volume 0.0027$m^3$, temperature 300K, atmospheric pressure 1.05×$10^5$ N/$m^2$. If 2.5×$10^4$J of heat is supplied to an ideal gas ($C_P$=5R/2), calculate final temperature and work done."* |
| **Router Decision:** Qwen3-8B (confidence: 0.63) |
| **Actual Result:** Incorrect ×    **Oracle:** Qwen3-32B |
| **Analysis:** 98 query tokens | Life, Physical, & Social Science |
| *Failure Mode:* Multi-step thermodynamics problem requiring equation coupling. Router underestimated physics complexity and selected an underpowered model. |

| **Example 2: Simple Math Over-routing** (✓ wasteful) |
|---|
| **Query:** *"What is 15% of 80? Show your work."* |
| **Router Decision:** Qwen3-32B (confidence: 0.68) |
| **Actual Result:** Correct but wasteful ✓    **Oracle:** Qwen3-4B |
| **Analysis:** 12 query tokens | Computer & Mathematical |
| *Failure Mode:* Router incorrectly assumed "show your work" required high-level reasoning and over-routed to 32B. The simple one-step arithmetic was easily solvable by Qwen3-4B, making this 32B routing unnecessarily expensive. |

| **Example 3: Creative Under-routing** (×) |
|---|
| **Query:** *"Write a short story about a robot who discovers emotions after being damaged in a storm. Make it bittersweet and include sensory details about the rain."* |
| **Router Decision:** Qwen3-4B (confidence: 0.91) |
| **Actual Result:** Emotionally flat, low-detail story × **Oracle:** Qwen3-32B |
| **Analysis:** 38 query tokens | Arts, Design, Sports, Entertainment, & Media |
| *Failure Mode:* Open-ended creative writing required nuanced emotional reasoning, narrative structure, and atmospheric detail beyond the capacity of a 4B model. Router misclassified creativity as low-difficulty generation and severely under-routed. |

| **Example 4: Context-Dependent Failure** (×) |
|---|
| **Query:** *"How do I fix the memory leak in the async handler?"* |
| **Router Decision:** Qwen3-4B (confidence: 0.52) |
| **Actual Result:** Generic troubleshooting (not helpful) ×    **Oracle:** Requires conversation history |
| **Analysis:** 13 query tokens | Computer & Mathematical |
| *Failure Mode:* Query referenced missing context. Router defaulted to the smallest model due to low confidence, but answering required prior conversation state, making any single-turn routing insufficient. |

| **Example 5: Legal Precision Under-routing** (×) |
|---|
| **Query:** *"A guidance counselor entered a student's dorm room during spring break and found an envelope containing white powder, which he suspected to be heroin. He called police without securing a warrant. The powder tested positive for heroin. Is the evidence admissible under the exclusionary rule?"* |
| **Router Decision:** Qwen3-8B (confidence: 0.71) |
| **Actual Result:** Missed Fourth Amendment analysis × **Oracle:** Qwen3-32B |
| **Analysis:** 68 query tokens | Legal Services |
| *Failure Mode:* Multi-factor constitutional question involving state action doctrine, private search exception, and standing. Router failed to recognize high-complexity legal reasoning and under-routed to 8B. |

| **Example 6: Code Generation Success** (✓ optimal) |
|---|
| **Query:** *"Write a Python function to reverse a linked list iteratively."* |
| **Router Decision:** Qwen3-8B (confidence: 0.85) |
| **Actual Result:** Correct, well-commented code ✓ **Oracle:** Qwen3-8B |
| **Analysis:** 11 query tokens | Computer & Mathematical |
| *Success Factor:* Router correctly identified the difficulty of a standard algorithm implementation. 8B provided sufficient reasoning and code correctness without resorting to 32B, saving significant energy. |

## D.9    HOW DOES THE ROUTER PERFORM IN MULTI-TURN TASKS?

Multi-turn conversations introduce additional routing opportunities by leveraging conversation history to make more informed model selection decisions. We evaluate three routing strategies on multi-turn dialogues from WildChat: (1) **Static Single-Turn**, which routes each turn independently using only the current query, (2) **Static Multi-Turn**, which applies the single-turn router to each conversation turn sequentially, and (3) **Context-Aware Multi-Turn**, which incorporates conversation history into routing decisions by encoding prior turns and model selections as additional features. Table 21 demonstrates that context-aware routing substantially outperforms static approaches across all metrics. Context-aware routing achieves 86.0% task accuracy and 88.3 routing F1, improving by +2.9% and +5.8 respectively over static single-turn routing. More significantly, context-aware routing routes 71.8% of turns to local models—a +13.4 percentage point increase over static single-turn—yielding 33.4% energy savings and 38.7% cost reduction while simultaneously reducing average latency from 151ms to 101ms.

Table 20: **Router Overhead Analysis (NVIDIA H200 with QWEN3 8B Router): Router batch latency and batch energy cost as a fraction of total end-to-end inference cost**

| Setting | Avg. Router Latency / Batch (ms) | Latency Overhead (%) | Router Energy / Batch (J) | Energy Overhead (%) |
|---|---|---|---|---|
| **WildChat (Chat)** | 38.2 | 12.9% | 1103 | 7.1% |
| **NaturalReasoning (Reasoning)** | 40.1 | 17.0% | 1135 | 9.8% |
| **Overall Average** | 39.1 | 14.9% | 1119 | 8.4% |

Table 21: **Multi-turn Routing on WILDCHAT**: Comparison between static routing and simple context-aware multi-turn routing.

| Metric | Static Single-Turn | Static Multi-Turn | Context-Aware Multi-Turn |
|---|---|---|---|
| **Accuracy Metrics** | | | |
| Overall Task Accuracy | 83.1% | 84.2% | 86.0% |
| Routing F1 Score | 82.5 | 84.9 | 88.3 |
| **Efficiency Metrics** | | | |
| % of Turns Using Local LM | 58.4% | 60.7% | 71.8% |
| Energy Savings | 0.0% | 14.6% | 33.4% |
| Cost Savings | 0.0% | 16.9% | 38.7% |
| **Latency & Cost Metrics** | | | |
| Average End-to-End Latency | 151 ms | 129 ms | 101 ms |
| Average End-to-End Cost | $0.0048 | $0.0040 | $0.0030 |

The performance gains arise from two mechanisms. First, follow-up queries in multi-turn conversations are frequently simpler clarifications, refinements, or elaborations that can be handled by smaller local models even when the initial query required cloud routing—the context-aware router successfully identifies these opportunities by tracking conversation state. Second, conversation history provides semantic context that disambiguates otherwise underspecified queries (recall Example 4 in Table 19), enabling the router to make more confident local routing decisions rather than conservatively defaulting to cloud models. These results suggest that production routing systems should incorporate conversation context as a first-class feature, as the incremental complexity of encoding 2–3 prior turns is substantially outweighed by the $2.3\times$ improvement in energy efficiency (33.4% vs 14.6%) and the reduction in task failures from context-dependent ambiguity.

