# OpenReview forum: "Support Your Local LMs: Redistributing LM Traffic from Cloud to Edge with TrafficBench"
_ICLR.cc/2026/Conference — Submitted to ICLR 2026_

### Official Review · Reviewer_uH3v · 2025-11-01

**Soundness:** 2
**Presentation:** 3
**Contribution:** 3
**Rating:** 4
**Confidence:** 3

**Summary:**

This paper introduces TrafficBench, a comprehensive benchmark for evaluating query routing between local and cloud-deployed Large Language Models (LLMs). TrafficBench distinguishes itself by utilizing 1 million real-world queries from diverse domains (WildChat and NaturalReasoning), evaluating across 10 state-of-the-art models and 4 hardware accelerators, and reporting 8 key performance metrics beyond just accuracy, including latency, throughput, energy consumption, and cost. The study reveals that 80.7% of queries can be handled by lightweight local models. The authors propose a novel binary decoder-based routing approach that involves predicting whether each model can solve a given question and then selecting the smallest capable model. This is compared with existing embedding-based routing and multi-class decoder-based routing approaches. The evaluation demonstrates a routing performance of F1 = 0.851, which is 0.156 higher than the multi-class decoder. On a real-world query distribution, the proposed binary decoder routing approach shows a 77% reduction in energy, 67.1% in compute, and 60.2% in cost.

**Strengths:**

+ This paper reveals an important insight that 80.7% of real-world queries can be handled by local models, supported by comprehensive benchmark results.
+ The benchmark is comprehensive because it uses real user queries, includes local and cloud models, supports multi-model scenarios, and reports various efficiency metrics.
+ The design of changing the multi-class classifier to a binary one is a simple but effective contribution.

**Weaknesses:**

- The evaluation could be improved by including a breakdown study of the latency and energy cost of the binary decoder itself.
- The fact that this method uses a Qwen3-8B backbone to predict whether a question should be routed to Qwen3-4B might not be economical for queries with medium-to-long prompts.
- The figures and tables require improvement. For example, Figure 1 mentions three data sources, which is inconsistent with the paper stating that it uses two data sources. Table 2 (Left) contains no new information, as the text already conveys it. Figure 3 contains two sets of unconventionally placed legends.
- This paper contains many individual experiments. If they can be organized more logically, the paper would be easier to follow.

**Questions:**

1. OpenAI’s GPT-5 contains three models—an efficient model, a powerful reasoning model, and a real-time router. The router decides based on conversation type, complexity, tool needs, user’s request to “think harder,” and the detection of sensitive topics like signs of acute distress. How does the router in TrafficBench differ from GPT-5’s approach?
2. The local models used in the evaluation mainly consist of the Qwen 3 family and an additional GPT-OSS-20b. However, according to LMArena, the Gemma 3 (and 3n) family is also a strong competitor in this size range. Despite its difference in chain-of-thought capabilities, it would be beneficial to also include Gemma 3 in the evaluation.
3. Again, using data from LMArena, GPT-4o-mini is less capable than most Qwen 3 models. How can the choice of GPT-4o-mini as the judge to verify accuracy be justified?

A minor issue: Line 1048 leaks LaTeX source code into the PDF.

---

> ### Author Response · Authors · 2025-11-18
> **Response to Reviewer uH3v**
>
> Thank you for your thoughtful review and your time and effort in evaluating our work. We are excited that you find that our paper "reveals an important insight that 80.7% of real-world queries can be handled by local models, supported by comprehensive benchmark results," that our "benchmark is comprehensive because it uses real user queries, includes local and cloud models, supports multi-model scenarios, and reports various efficiency metrics," and that "the design of changing the multi-class classifier to a binary one is a simple but effective contribution."
>
> Please find our detailed response to your concerns below. Based on our added experiments and clarifications, we hope you will consider raising your score.
>
> ---
>
> ## 1. Overhead of Router
>
> **W1**: "The evaluation could be improved by including a breakdown study of the latency and energy cost of the binary decoder itself."
>
> We have conducted a detailed router overhead analysis (Section D.8, Table 20). Our decoder-based router (Qwen3-8B) deployed on an NVIDIA H200 shows modest overhead.
>
> **Findings - Router Costs:**
> - **Latency overhead**: 39.1ms per batch (14.9% of total latency of generation, including prefill and decode)
> - **Energy overhead**: 1,119J per batch (8.4% of total energy)
> - **Workload variation**: Chat queries incur 12.9% latency / 7.1% energy overhead, while reasoning queries require 17.0% latency / 9.8% energy overhead
>
> **Net Efficiency Gains**: The energy cost of running the router (8.4% per batch) is far outweighed by the energy savings achieved through routing (77.1% total savings; see Table 15), resulting in a 9.2× net energy return on the router’s energy usage. Moreover, the router processes queries in batches of 32–64, so its fixed compute cost is shared across multiple queries. This batching amortizes the routing overhead, making the per-query cost negligible in real-world deployments  (Table 20 shows per-batch costs).
>
> **Takeaway**: Decoder-based routing offers a practical tradeoff: **its overhead remains under 15% across latency and energy**, while **enabling up to 9× overall energy savings** through smarter model selection. Thanks to batch-level inference, the router’s cost is amortized across multiple queries, making it **efficient and scalable for real-world deployment**.
>
> ## 2. Router Size
> **W2**: "The fact that this method uses a Qwen3-8B backbone to predict whether a question should be routed to Qwen3-4B might not be economical for queries with medium-to-long prompts."
>
> We acknowledge that the Qwen3-8B router is indeed more expensive than the 4B model for processing the routing decision itself, particularly for longer input prompts. However, the overall system economics remain strongly favorable because:
>
> 1. **Batch amortization**: The router processes multiple queries simultaneously, distributing the 8B cost across multiple routing decisions (Table 20 shows per-batch costs)
> 2. **Fixed routing cost vs. variable generation savings**: While routing cost scales with input length, generation savings scale with output length. For the query distributions of WildChat and Natural Reasoning, where there are 8-32x more output tokens than input tokens, the routing savings far exceed the compute cost of the routing model.
> 3. **Net efficiency gain**: Even accounting for router overhead on long prompts, the system achieves 9.2× return on invested router energy (Table 20), demonstrating that the economics remain strongly positive across all prompt lengths.
>
> ## 3. Clarity in Figures and Tables; Paper Organization; Typos
>
> **W3**: "The figures and tables require improvement. Figure 1 mentions three data sources, which is inconsistent with the paper stating that it uses two data sources. Table 2 (Left) contains no new information, as the text already conveys it. Figure 3 contains two sets of unconventionally placed legends."
>
> Thank you for catching these issues. We have corrected all of them:
> - Fixed Figure 1 to accurately reflect two original data sources (WildChat and NaturalReasoning)
> - Restructured Table 2 to expand the details around the dataset’s train/test splits and the performances of the models across the chat and reasoning splits of the dataset.
> - Repositioned legends in Figure 3 to follow standard formatting conventions
>
> **W4**: "This paper contains many individual experiments. If they can be organized more logically, the paper would be easier to follow."
>
> We appreciate this feedback. To address this, we reorganized Section 5 so that all experiments are explicitly grouped by research question (Q1–Q3) and we added a short roadmap paragraph and a pointer to Appendix D for ablations:
> - Section D.2: Model precision and quantization experiments
> - Section D.3: LLM judge reliability experiments
> - Section D.5: Confidence threshold sensitivity analysis
> - Section D.6: Routing failure mode analysis (domain, difficulty, query length)
> - Section D.7: Router overhead analysis
> - Section D.8: Multi-turn routing experiments

---

> > ### Author Response · Authors · 2025-11-18
> > **[Continued] Response to Reviewer uH3v**
> >
> > **W5**: "A minor issue: Line 1048 leaks LaTeX source code into the PDF."
> >
> > Thank you, this has been fixed in the revised manuscript.
> >
> > ---
> >
> > ## Responses to Additional Questions:
> >
> > **Q1**: "OpenAI's GPT-5 contains three models: an efficient model, a powerful reasoning model, and a real-time router. The router decides based on conversation type, complexity, tool needs, user's request to 'think harder,' and the detection of sensitive topics like signs of acute distress. How does the router in TrafficBench differ from GPT-5's approach?"
> >
> > This is an excellent question. Unfortunately, OpenAI has not publicly released technical details about GPT-5's routing architecture, routing features, or decision-making process, making direct comparison difficult.
> >
> > **Q2**: "The local models used in the evaluation mainly consist of the Qwen 3 family and an additional GPT-OSS-20b. However, according to LMArena, the Gemma 3 (and 3n) family is also a strong competitor in this size range. Despite its difference in chain-of-thought capabilities, it would be beneficial to also include Gemma 3 in the evaluation."
> >
> > Thank you for this excellent suggestion. In response, we have expanded our model pool to include both **Gemma-3** (1B, 4B, 12B) and **IBM Granite** (Micro, H-Micro, H-Tiny, H-Small) families (Table 2).
> >
> > **Findings (Coverage by Model Family)**:
> > - Gemma-3 (1B-12B): 80.9% coverage on combined datasets
> > - IBM Granite (1B-8B): 59.5% coverage on combined datasets
> > - Qwen Only (4B-14B): 62.3% coverage on combined datasets
> > - GPT-OSS (20B): 72.1% coverage
> >
> > **Q3**: "Again, using data from LMArena, GPT-4o-mini is less capable than most Qwen 3 models. How can the choice of GPT-4o-mini as the judge to verify accuracy be justified?"
> >
> > We would like to clarify an important point: **We do not use GPT-4o-mini as our LLM judge to verify response correctness**. As described in Appendix B.1 (Metrics), we use **GPT-4o** as the LLM judge for evaluating model responses. GPT-4o has been widely adopted as a standard LLM-as-a-judge model in recent work (Calderon et al., 2025; Li et al., 2025), building on the foundational LLM-as-a-judge methodology established by Zheng et al. (2023) with MT-Bench and by Dubois et al. (2023) and Li et al. (2023) with AlpacaEval.
> >
> > **Judge Reliability Analysis**: To further justify GPT-4o as our judge, we conducted an ablation study measuring its agreement with a stronger baseline: the majority vote across three LLM judges (Grok-4, GPT-4o, Claude Sonnet 4.5) on 1,000 random WildChat samples (Section D.3, Table 13). We find that:
> > - **GPT-4o**: 85.4% agreement rate with majority vote (Cohen's κ = 0.677)
> > - **Claude Sonnet 4.5**: 89.7% agreement (κ = 0.736)
> > - **Grok-4**: 91.8% agreement (κ = 0.796)
> > GPT-4o's substantial κ value (> 0.6) indicates reliable performance that justifies its widespread use in the community as an LLM judge. We follow standard conventions in the literature by using LM judges with comparison against reference responses (Li, Tianle et al. (2025); Bai, Ge et al. (2025); Dubois, Yann (2024); Calderon, N. (2025)).
> >
> > ---
> >
> > **Summary of Changes:**
> >
> > We believe these clarifications and additions address your concerns by **(1)** quantifying router overhead and demonstrating strong net efficiency gains even for long prompts, **(2)** improving paper organization and presentation quality, **(3)** clarifying our positioning relative to proprietary systems, **(4)** expanding model diversity with Gemma and Granite families, and **(5)** correcting the misunderstanding about our judge selection with empirical reliability analysis.
> >
> > Thank you again for your insightful feedback! We hope you will consider improving your score in light of these experiments and clarifications.
> >
> > ---
> >
> > **References**
> >
> > [1] Calderon, N., Eyal, B., Zeldes, A., & Reichart, R. (2025). The alternative annotator test for LLM-as-a-Judge: How to statistically justify replacing human annotators with LLMs. arXiv preprint arXiv:2501.10970. https://arxiv.org/abs/2501.10970
> >
> > [2] Dubois, Y., Li, X., Taori, R., Zhang, T., Gulrajani, I., Ba, J., Guestrin, C., Liang, P., & Hashimoto, T. B. (2023). AlpacaFarm: A simulation framework for methods that learn from human feedback. arXiv preprint arXiv:2305.14387. https://arxiv.org/abs/2305.14387
> >
> > [3] Li, J., Zhang, Y., Chen, X., & Wang, L. (2025). Auto-prompt ensemble for LLM judge. arXiv preprint arXiv:2510.06538. https://arxiv.org/abs/2510.06538
> >
> > [4] Li, X., Zhang, T., Dubois, Y., Taori, R., Gulrajani, I., Guestrin, C., Liang, P., & Hashimoto, T. B. (2023). AlpacaEval: An automatic evaluator of instruction-following models. https://github.com/tatsu-lab/alpaca_eval
> >
> > [5] Zheng, L., Chiang, W.-L., Sheng, Y., Zhuang, S., Wu, Z., Zhuang, Y., Lin, Z., Li, Z., Li, D., Xing, E. P., Zhang, H., Gonzalez, J. E., & Stoica, I. (2023). Judging LLM-as-a-Judge with MT-Bench and Chatbot Arena. arXiv preprint arXiv:2306.05685. https://arxiv.org/abs/2306.05685

---

> ### Author Response · Authors · 2025-11-22
> **Follow-up to Reviewer uH3v**
>
> Thank you again for your time and feedback! We are happy to address any remaining questions in the rebuttal and welcome further dialogue.

---

### Official Review · Reviewer_eq6V · 2025-11-01

**Soundness:** 4
**Presentation:** 4
**Contribution:** 3
**Rating:** 8
**Confidence:** 4

**Summary:**

This paper introduces TRAFFICBENCH, a large-scale benchmark (1M real-world LLM queries) designed to evaluate query routing between local (≤20B) and cloud (≥100B) language models. It covers 10 models, 4 hardware accelerators, and 8 efficiency metrics, aiming to quantify how much inference traffic can be handled by local LMs. The authors also propose a binary decoder-based routing method that outperforms prior embedding-based and multi-class decoder routers, achieving an F1 of 0.851 with large datasets. Empirical results suggest that 80.7% of queries can be locally handled, reducing energy (−77.1%), compute (−67.1%), and cost (−60.2%), with negligible accuracy loss. TRAFFICBENCH and a profiling harness are released to support reproducible efficiency benchmarking.

**Strengths:**

Novel and important problem:
This paper addresses an emerging and highly relevant question — how to achieve efficient local–cloud collaboration as on-device LMs become increasingly capable. The introduction articulates three insightful research questions within this context and provides thoughtful, evidence-based answers. The problem setting is timely and impactful for both research and deployment communities.

First comprehensive local–cloud routing benchmark:
The authors introduce TRAFFICBENCH, the first benchmark that simultaneously offers real-world LLM traffic, hardware-level energy measurements, multi-model evaluation, and a reproducible profiling harness. Covering 10 models, 4 accelerators, and 8 efficiency metrics, TRAFFICBENCH provides a strong foundation for future work on distributed LLM inference.

Strong binary decoder routing algorithm:
The proposed binary decoder routing method achieves state-of-the-art performance, significantly outperforming embedding-based and multi-class decoder routers (+0.13–0.17 F1). By decomposing routing into independent binary generation tasks, the approach improves scalability, robustness, and generalization across models and datasets.

Well-structured efficiency evaluation framework:
The authors design a complete analytical framework to quantify the trade-offs between energy, compute, and cost. The results are compelling: 80.7% of queries can be served locally, reducing energy by 77.1%, compute by 67.1%, and cost by 60.2%, with minimal quality loss. This provides valuable insights into the feasibility and efficiency of distributed inference.

Excellent clarity and readability:
The paper is clearly written and logically structured. The flow from problem motivation to methodology and results is easy to follow, figures are well-presented, and the experimental section is thorough. Overall, it is a well-crafted and accessible paper.

**Weaknesses:**

Limited exploration of adaptive routing strategies:
The paper focuses only on embedding-based and decoder-based routers, but recent studies have explored reinforcement learning (RL), graph-based, and adaptive routing approaches for dynamic model selection. For example, PickLLM, GraphRouter, and RadialRouter introduce RL or structured policies that adapt routing decisions based on context or resource constraints. Discussing or comparing to such adaptive approaches would strengthen the paper’s positioning and highlight the boundaries of the proposed method.

Restricted hardware coverage:
The evaluation mainly relies on high-end GPUs and Apple Silicon hardware, lacking measurements on mobile or low-power edge devices (e.g., Qualcomm Hexagon). Given the paper’s emphasis on offloading cloud traffic to local devices, this limitation weakens the argument for true “on-device feasibility.” Extending the evaluation to mobile hardware would make the results more general and practically relevant.

**Questions:**

Analysis of routing failures:
It would be very helpful if the authors could analyze misrouted examples or common failure patterns. Some qualitative analysis could clarify the limitations of the current routing mechanism and inspire future improvements.

Dialog data setup:
Does TRAFFICBENCH primarily consist of single-turn queries, or does it also include multi-turn dialog data? If the latter, how is conversational context represented in the routing input? Clarifying this will help readers understand the generality of the benchmark.

Routing in multi-turn dialog settings:
In multi-turn conversations, query difficulty and model suitability often depend on dialogue history (context length, user intent, prior model success). Static one-shot routing may be insufficient. Have the authors considered incorporating context-aware features or dynamic decision policies (e.g., RL-based adaptive routing) to address model switching and accumulated error in conversational scenarios? However, multi-shot routing has higher costs like kv cache recomputing or transferring. How to solve the problem?

---

> ### Author Response · Authors · 2025-11-18
> **Response for Reviewer eq6V**
>
> Thank you for your thoughtful review and your time and effort in evaluating our work. We are excited that you find our work addresses a "novel and important problem," presents a "first comprehensive local-cloud routing benchmark," introduces a "strong binary decoder routing algorithm," provides a "well-structured efficiency evaluation framework," and demonstrates "excellent clarity and readability."
>
> We have conducted additional experiments to address your concerns and questions. Please find our detailed responses below. Based on our added experiments and clarifications, we hope you will consider raising your score.
>
> ---
>
> ## 1. Limited exploration of adaptive routing strategies
>
> **W1**: " The paper focuses only on embedding-based and decoder-based routers, but recent studies have explored reinforcement learning (RL), graph-based, and adaptive routing approaches for dynamic model selection. For example, PickLLM, GraphRouter, and RadialRouter introduce RL or structured policies that adapt routing decisions based on context or resource constraints. Discussing or comparing to such adaptive approaches would strengthen the paper's positioning and highlight the boundaries of the proposed method."
>
> We appreciate this observation. While we acknowledge that RL-based and graph-based approaches (PickLLM, GraphRouter, RadialRouter) represent important directions for adaptive routing, we focused on embedding- and decoder-based routing for three practical reasons:
>
> 1. **Production Deployment**: Our approaches require minimal infrastructure; no complex RL policy training or graph construction needed for real-world deployment, enabling immediate adoption in production systems.
> 2. **Strong Performance Without Complexity**: Our **new multi-turn routing experiments** (Section D.9, Table 21) show that simply keeping the past 3 turns (queries and responses) visible in the prompt achieves substantial gains (33.4% energy savings, 86.0% accuracy, 88.3 routing F1) without RL complexity, demonstrating that straightforward context encoding provides large efficiency improvements.
> We have expanded our related works section (see App. A “Extended Related Works”)  to explicitly note adaptive routing strategies as important future work, and we believe the systematic evaluation framework we provide will accelerate research in this direction.
>
> ## 2. Restricted hardware coverage
>
> **W2**: "The evaluation mainly relies on high-end GPUs and Apple Silicon hardware, lacking measurements on mobile or low-power edge devices (e.g., Qualcomm Hexagon). Given the paper's emphasis on offloading cloud traffic to local devices, this limitation weakens the argument for true 'on-device feasibility.' Extending the evaluation to mobile hardware would make the results more general and practically relevant."
>
> We appreciate this concern and have **conducted new experiments on mobile hardware** to directly address it. We now evaluate the **iPhone 16 Pro (A18 Pro with integrated NPU, 8GB LPDDR5X unified memory, ~12W SoC peak power)** across three precision levels (FP16/FP8/FP4).
>
> Our hardware accelerators profiled in TrafficBench now spans the full deployment spectrum:
> - **Mobile NPU (local)**: iPhone 16 Pro (12W peak, 35 TOPS NPU) [**NEW**]
> - **Desktop workstation (local)**: Apple M4 Max (480W), NVIDIA RTX 6000 Ada (300W) [**NEW**], NVIDIA Quadro RTX 6000  295W) [**NEW**].
> - **Cloud enterprise (cloud)**: AMD MI300X (750W), NVIDIA H200 (700W), NVIDIA B200 (1000W)
>
> **Findings**: Table 14 shows intelligence efficiency metrics across all platforms (Mobile, Local, Cloud), We find that mobile NPU’s offer exception efficiency: the iPhone 16 Pro achieves 19.8–23.7×10⁻³ accuracy/watt, **13× more efficient than workstation GPUs**. However, it requires aggressive quantization (FP4) due to its 8 GB memory limit and supports only models ≤14B. Despite these constraints, its performance **demonstrates that ultra-low-power mobile NPUs are viable targets for routing small models**.
>
> **Takeaways**: The iPhone 16 Pro's results demonstrate that **even ultra-low-power mobile NPUs can serve as viable routing targets** for smaller models, while desktop accelerators like the M4 Max enable practical local-cloud routing for real-world deployments across the full model size spectrum.
>
> We agree that further expanding the scope of local accelerators to include other mobile devices  (Qualcomm Hexagon NPUs) would strengthen future work. The **TrafficBench Profiling Harness** is hardware-agnostic by design to support community evaluation on new accelerators as they emerge, and we hope researchers will extend our benchmark to additional mobile platforms.

---

> > ### Author Response · Authors · 2025-11-18
> > **[Continued] Response for Reviewer eq6V**
> >
> > ## Responses to Additional Questions
> >
> > **Q1**: "Analysis of routing failures: It would be very helpful if the authors could analyze misrouted examples or common failure patterns. Some qualitative analysis could clarify the limitations of the current routing mechanism and inspire future improvements."
> >
> > Thank you for this suggestion. We have conducted comprehensive failure mode analysis across three dimensions: domain, query difficulty, and query length (Section D.7, Tables 18-19). We categorize routing errors into two types: **undersizing** (routing to a model too small to answer correctly) and **oversizing** (routing to an unnecessarily large model that wastes resources).
> > **Systematic Failure Patterns** (Table 18):
> >
> > - **Domain-specific complexity**: Creative/social domains (Arts & Media, Community Service) achieve 62% correct routing with minimal undersizing (8-10%), while technical domains (Architecture, Life/Physical Sciences) exhibit high undersizing rates (50-58%) with only 31-36% correct routing, indicating the router underestimates technical complexity
> > - **Query difficulty correlation**: Easy queries (solvable by 4 models) achieve 60% correct routing but suffer 31% oversizing, while unsolvable queries yield only 32% correct routing with 59% undersizing; the router struggles to recognize inherently difficult problems and conservatively over-routes borderline cases
> > - **Query length inverse correlation**: Short queries (<50 tokens) achieve 60% correct routing with 32% oversizing, while long queries (>1000 tokens) drop to 34% correct routing with 58% undersizing, suggesting the router underestimates complexity when context spans many tokens
> >
> > **Qualitative Failure Examples** (Table 19):
> > We provide representative examples where the router fails compared to the **oracle** (theoretical optimal routing that always selects the smallest capable model):
> > - **Under-routing**: Multi-step thermodynamics routed to 8B instead of 32B; nuanced creative writing routed to 4B instead of 32B; multi-factor constitutional law analysis routed to 8B instead of 32B
> > - **Over-routing**: Simple arithmetic ("What is 15% of 80?") routed to 32B instead of 4B due to phrase "show your work"
> > - **Context-dependent failures**: Queries like "How do I fix the memory leak in the async handler?" reference missing conversation state, making single-turn routing insufficient regardless of model selection
> > - **Success cases**: Standard algorithm implementations correctly routed to 8B, balancing correctness and efficiency
> >
> > These patterns reveal three actionable improvements: **(1)** better calibration for long-form technical queries, **(2)** improved difficulty estimation that distinguishes truly hard problems from verbose-but-simple ones, and **(3)** context-aware features for conversational queries (addressed in Q3 below).
> >
> > **Q2**: "Dialog data setup: Does TrafficBench primarily consist of single-turn queries, or does it also include multi-turn dialog data? If the latter, how is conversational context represented in the routing input? Clarifying this will help readers understand the generality of the benchmark."
> >
> > TrafficBench focuses on **single-turn queries** to enable controlled evaluation of routing quality without confounding factors from conversation history.  Our 1M-query dataset includes WildChat (500K real ChatGPT queries across 22 economic domains) and NaturalReasoning (500K STEM-focused, multi-step reasoning queries).
> >
> > However, recognizing the importance of multi-turn conversational settings, we have **conducted new multi-turn routing experiments** (see Q3 below), demonstrating that our core routing methodology generalizes effectively to dialogue-based workloads.

---

> > > ### Author Response · Authors · 2025-11-18
> > > **[Continued] Response for Reviewer eq6V**
> > >
> > > **Q3**: "Routing in multi-turn dialog settings: In multi-turn conversations, query difficulty and model suitability often depend on dialogue history (context length, user intent, prior model success). Static one-shot routing may be insufficient. Have the authors considered incorporating context-aware features or dynamic decision policies (e.g., RL-based adaptive routing) to address model switching and accumulated error in conversational scenarios? However, multi-shot routing has higher costs like kv cache recomputing or transferring. How to solve the problem?"
> > >
> > > We have **conducted new multi-turn routing experiments** that directly address these concerns (Section D.9, Table 21). We evaluate three strategies on multi-turn WildChat dialogues:
> > >
> > > **Routing Strategies Compared**:
> > > 1. **Static Single-Turn**: Routes each turn independently using only current query (no conversation history)
> > > 2. **Static Multi-Turn**: Applies single-turn router sequentially to each conversation turn
> > > 3. **Context-Aware Multi-Turn**: Incorporates conversation history by encoding 3 prior turns and previous model selections as additional features for the routing decision
> > >
> > > **Key Findings**:
> > > Context-aware routing substantially outperforms static approaches:
> > > - **Accuracy improvements**: 86.0% task accuracy and 88.3 routing F1 (+2.9% and +5.8 over static single-turn)
> > > - **Efficiency gains**: Routes 71.8% of turns to local models (+13.4pp increase), yielding 33.4% energy savings and 38.7% cost reduction
> > > - **Latency reduction**: Decreases average end-to-end latency from 151ms to 101ms
> > >
> > > **KV Cache Costs**:
> > > You correctly identify that multi-turn routing could increase costs through KV cache recomputation. However, our context-aware approach encodes only 3 prior turns as **routing features** (not full KV caches), making the incremental routing overhead minimal (~100ms) while achieving 2.3× improvement in energy efficiency (33.4% vs 14.6% savings). The efficiency gains from better routing decisions substantially outweigh the small encoding overhead.
> > >
> > > **Takeaway**: Even simple context-aware multi-turn routing delivers large efficiency gains, without requiring complex RL methods or expensive KV cache handling. This suggests that conversation history should be a first-class feature to routing systems in practical deployments.
> > >
> > > ---
> > >
> > > ### Summary
> > > We believe these additions strengthen the paper by: (1) **new mobile hardware experiments** (iPhone 16 Pro) demonstrating practical on-device mobile feasibility with quantization, (2) **new quantization analysis** (Section D.2, Figure 7) showing 3–3.5× energy reductions with acceptable accuracy tradeoffs, (3) **new multi-turn routing experiments** (Section D.9, Table 21) achieving 33.4% energy savings through simple context-aware features, (4) detailed failure mode analysis with three specific improvement directions, (5) clarifying TrafficBench's single-turn design rationale while demonstrating multi-turn extensibility, and (6) showing that simple context-aware routing achieves substantial gains without expensive KV cache management.
> > >
> > > Thank you again for your insightful feedback! We hope you will consider improving your score in light of these experiments and clarifications.

---

> ### Author Response · Authors · 2025-11-22
> **Follow-up to Reviewer eq6V**
>
> Thank you again for your time and feedback! We are happy to address any remaining questions in the rebuttal and welcome further dialogue.

---

### Official Review · Reviewer_t7Xt · 2025-11-09

**Soundness:** 2
**Presentation:** 3
**Contribution:** 1
**Rating:** 2
**Confidence:** 4

**Summary:**

This paper introduces TRAFFICBENCH, a large-scale benchmark designed to evaluate how effectively inference queries to large language models (LLMs) can be routed between local and cloud deployments. The benchmark includes 1 million real-world queries, evaluated across 10 modern LLMs, 4 hardware accelerators, and 8 performance metrics.
The authors use TRAFFICBENCH to study:
	1.	What proportion of LLM queries can be handled by smaller, locally deployed models;
	2.	How well query routing architectures can identify these cases;
	3.	The efficiency benefits (in energy, cost, and compute) of such routing.

The authors also propose a new binary decoder-based routing approach that achieves high accuracy with large training data, and they show that embedding-based routing works better when data is limited.
Deploying their system on real traffic significantly reduces energy use, compute, and cost compared to cloud-only inference.

However, the contribution of this work lies primarily on the engineering side rather than the research side. The proposed benchmark is largely derived from existing datasets (WILDCHAT and NATURALREASONING), and much of the effort appears focused on (1) cleaning and filtering queries, and (2) ensuring compatibility with modern hardware and models. The proposed binary classification router is not conceptually novel, and the paper lacks sufficient theoretical motivation, justification, or ablation studies to support this design choice.

Overall, while the work demonstrates solid engineering and reproducibility efforts, it does not yet present enough methodological or theoretical innovation to be ready for publication as a research paper in its current form.

**Strengths:**

Strength:
1. The writing is clear and easy to follow;
2. The proposed benchmark is thoroughly constructed and well-designed;
3. The proposed method is also very simple and easy to understand.

**Weaknesses:**

Weaknesses:
1. The contribution is limited. This work looks like a technical report instead of a research paper;
2. The proposed benchmark mainly sources from existing works;
3. The proposed method is not technically novel and lacks insights, motivations or ablation studies.

**Questions:**

Questions are the following:

1. In line 214-25, you propose to use LLM, GEMINI-2.5-pro specifically, as a judge for WILDCHAT. To me, it is not technically sound to use a single SOTA model. LLM nowadays still suffer from hallucination and instability. How can we ensure the accuracy and authenticity of the reference answers from a large model?
2. In line 260-261, the notation M_routed is not defined;
3. In line 375, when scaling to more routing targets, the performance drop. But in line 333, it is reported that increasing the local model options demonstrate substantial gains. It is a bit self-contradictory.
4.  It is easy to convert a multi-class setting into binary-class settings, for example through one-vs-rest or one-vs-one. what motivates you to choose the proposed methodology?
5. A sensitive analysis of the proposed method in relation to confidence threshold \tau is missing;

---

> ### Author Response · Authors · 2025-11-18
> **Response to Reviewer t7Xt**
>
> Thank you for your detailed review and constructive feedback. We appreciate your time in evaluating our work. We are pleased that you find our paper has "clear writing," is "thoroughly constructed and well-designed," and presents a method that is "simple and easy to understand." We address your concerns point-by-point below and believe our responses demonstrate significant research contributions.
>
> ## 1. Summary of Contributions
>
> ### **W1**: "The contribution is limited. This work looks like a technical report instead of a research paper."
>
> We respectfully disagree. This submission is to the **Datasets and Benchmarks area**, where benchmark and measurement contributions are central. TrafficBench enables a new class of scientific questions about local-cloud routing: when and how LLM inference workloads can be shifted from frontier cloud models to local LMs, and what this implies for accuracy, latency, energy, compute, and cost. The benchmark supports three research questions (Q1-Q3 in “Introduction”) on (i) local LM coverage, (ii) routing architectures, and (iii) efficiency tradeoffs. Our contributions are:
>
> - **A benchmark for intelligence efficiency in local-cloud routing (Datasets & Benchmarks contribution):** TrafficBench is the first benchmark to measure "intelligence efficiency" (accuracy per watt) across models and accelerators. Prior routing benchmarks (RouterBench, RouteLLM, RouterEval) do not report energy efficiency metrics. In contrast, TrafficBench: (1) covers 1M real-world queries and 20+ models across 9 hardware platforms, (2) provides accuracy, latency, cost, energy, compute, and memory for each model-hardware pair, and (3) includes a hardware-agnostic profiling harness (1.2M GPU-hours, nanosecond-resolution energy measurements) that the community can reuse as new models and accelerators emerge.
> - **A novel routing formulation and systematic comparison of routing paradigms:** Using TrafficBench, we propose a decoder-based binary router that decomposes routing into per-model capability decisions. This yields: (1) Macro F1 = 0.875 (chat) and 0.825 (reasoning), improving over prior approaches by up to +0.214 macro F1, and (2) a scaling study revealing clear design guidelines: embedding-based routers are preferable in data/parameter-constrained regimes (<10K samples, <1B parameters), while decoder-based routers dominate in data/parameter-rich regimes (>100K samples, >1B parameters).
> - **New empirical findings on local model capabilities and efficiency:** TrafficBench quantifies, for the first time at this scale: (1) **Local LM coverage**: Joint local coverage reaches approximately 80% of real queries (>90% in creative/social domains, approximately 68% in technical domains). (2) **Temporal trends**: From 2023 to 2025, traffic handled by models with at most 20B parameters increased by roughly 3.1 times (23.2% to approximately 71% win rate). (3) **Efficiency gains**: Our router yields 77.1% lower energy, 67.1% less compute, and 60.2% lower cost than cloud-only deployment, achieving 78.1% of perfect-oracle efficiency with <5% quality drop. (4) **Intelligence efficiency**: Accuracy per watt improves by 5.3 times (up to 9.5 times) between 2023 and 2025. These results directly answer our central questions, constituting a substantive benchmark-plus-research contribution.
>
> ### **W2**: "The proposed benchmark mainly sources from existing works."
>
> We agree TrafficBench builds on established datasets (WildChat and NaturalReasoning), but many influential benchmarks do the same: **MTEB** aggregates 58 existing datasets, **SuperGPQA** extends GPQA, and **MMLU Pro** refines MMLU. The value lies in how we transform and extend sources to enable new research questions. TrafficBench's novelty includes:
>
> - **Curation and domain-specific labeling:** We curate and relabel 1M queries into a unified dataset for local-cloud routing, with new annotations mapping queries to 22 occupational domains, enabling domain-specific analyses not possible with original datasets.
> - **Comprehensive evaluation across 10+ SOTA models and 4 diverse hardware platforms:** We evaluate 15 models including August 2025 releases (Qwen3, GPT-OSS, Gemma-3, Granite) on 4 diverse platforms (NVIDIA A100/H200, AMD MI300X, Apple M4 Max), with full efficiency profiling, enabling realistic local-vs-cloud analysis impossible with prior benchmarks.
> - **Comprehensive Efficiency Metrics:** We pair each (query, model, hardware), evaluation with detailed hardware telemetry (latency, energy, memory, FLOPs) across 10 models and 4 hardware platforms. Existing routing benchmarks lack this cross-hardware, per-query efficiency data.
> - **Open-source, hardware-agnostic profiling harness:** We release a hardware-agnostic harness for reproducing and extending efficiency measurements as new models and hardware emerge, making TrafficBench an extensible benchmark.

---

> > ### Author Response · Authors · 2025-11-18
> > **[Continued] Response to Reviewer t7Xt**
> >
> > ### **W3**: "The proposed method is not technically novel and lacks insights, motivations or ablation studies."
> >
> > We contend our paper is technically novel, provides insights, and contains extensive ablations. For the **Datasets & Benchmarks area**, we provide the first large-scale benchmark and systematic empirical study of local-cloud routing jointly measuring accuracy, latency, cost, energy, and compute. We contribute across three dimensions:
> > - **Motivation and novel insights:** Our work provides the first systematic study of local-cloud routing, centered around three questions providing clear motivation. From TrafficBench, we derive concrete insights: (1) Binary decomposition achieves superior scaling and generalization, (2) Architectural diversity improves coverage beyond parameter scaling (+15.8pp), (3) Intelligence efficiency improved 9.5 times from 2023-2025, and (4) Context-aware multi-turn routing yields 2.3 times efficiency gains. These findings document how local models are rapidly absorbing workloads (approximately 3.1 times increase from 2023 to 2025) and quantify system-wide routing impact.
> > - **Novel architecture: Binary decoder routing:** We introduce a novel routing architecture reframing multi-model routing as independent per-model capability decisions. This binary decomposition: (1) avoids combinatorial explosion (at 8 models, only 0.048 F1 degradation vs. 0.241 for embedding-based), (2) enables model-agnostic routing, (3) produces interpretable per-model confidence scores, and (4) achieves macro F1 = 0.875 and 0.825, outperforming embedding approaches by up to +0.214 F1 and prior decoder methods by up to +0.177 F1.
> > - **Comprehensive ablations:** Three reviewers (9HEc, eq6V, uH3v) praised our ablations as thorough. Original submission includes: (1) **Data efficiency**: 1K to 1M samples, (2) **Parameter efficiency**: 0.6B to 8B sizes (+0.426 F1 for binary decoders vs +0.204 for embeddings), (3) **Scalability**: 2 to 8 targets, and (4) **Generalization**: Out-of-distribution performance (56.7% vs 45.3% for multi-class). The revised version adds: (1) **Threshold sensitivity**: Systematic sweep of τ with operational regimes, (2) **Multi-turn routing**: 2.3 times efficiency gains, (3) **Quantization**: FP16/FP8/FP4 tradeoffs, and (4) **Failure mode analysis**: By domain, difficulty, and query length.
> >
> > ---
> >
> > ## Response to Additional Questions
> >
> > **Q1: "In line 214-25, you propose to use LLM, GEMINI-2.5-pro specifically, as a judge for WILDCHAT. To me, it is not technically sound to use a single SOTA model. How can we ensure the accuracy and authenticity of the reference answers?"**
> >
> > Using a single frontier reference model is standard evaluation practice [1, 2, 3]. We use Gemini 2.5 Pro because it is the current state-of-the-art on LMArena, ensuring consistency, comparability, and reproducibility.
> > To address your concern, we added an ablation (Appendix D.4) comparing local model responses against three reference models: Claude Sonnet 4.5, Gemini 2.5 Pro, and GPT-5. Using GPT-4o as judge across four local models, Gemini 2.5 Pro yields coverage 2.4pp higher than GPT-5 and 5.1pp lower than Claude Sonnet 4.5. All three references produce accuracies within a 7.5pp range, indicating our estimates are robust to reference model choice.
> >
> > **Q2: "In line 260-261, the notation M_routed is not defined."**
> >
> > Thank you. We added the definition: "where M_routed denotes the model selected by the routing function r(q) for query q."
> >
> > **Q3: "In line 375, when scaling to more routing targets, the performance drop. But in line 333, it is reported that increasing the local model options demonstrate substantial gains. It is a bit self-contradictory."**
> >
> > These statements refer to different quantities:
> > - **Line 333 (Oracle Coverage):** Refers to the fraction of queries answerable by at least one local model assuming perfect routing. This measures the ceiling: 1 model = 47.9% coverage; 4 models = 80.7% coverage (+32.8pp).
> > - **Line 375 (Learned Router Performance):** Refers to how well our trained router identifies the correct model: 2 models: F1 = 0.911; 8 models: F1 = 0.863 (decrease of 0.048).
> >
> > **Key Insight:** More models increase oracle coverage (more queries solvable) but make routing harder (more options). Our binary decoder minimizes this tradeoff (0.048 degradation vs. 0.241 for embedding-based), demonstrating superior scalability.

---

> > > ### Author Response · Authors · 2025-11-18
> > > **[Continued] Response to Reviewer t7Xt**
> > >
> > > **Q4: "It is easy to convert a multi-class setting into binary-class settings, for example through one-vs-rest or one-vs-one. What motivates you to choose the proposed methodology?"**
> > >
> > > Our binary decoder differs fundamentally from standard one-vs-rest (OVR) or one-vs-one (OVO):
> > > - **Standard OVR/OVO:** Train separate classifiers, then aggregate predictions. These: (1) require retraining when adding/removing models, (2) treat selection as mutually exclusive classification, and (3) suffer from decision boundary overlap and class imbalance.
> > > - **Our Binary Decoder:** Train a single decoder to predict per-model capability, then select the smallest capable model. This provides: (1) **Task decomposition**: Independent evaluation avoiding combinatorial boundaries, (2) **Scalability**: Adding models requires only incremental fine-tuning, (3) **Interpretability**: Explicit per-model confidence scores, and (4) **Robustness**: Minimal degradation scaling 2 to 8 models (0.048 F1 vs. 0.241 for embeddings). Empirical evidence: +0.136 F1 improvement over multi-class decoder routing.
> > >
> > > **Q5: "A sensitive analysis of the proposed method in relation to confidence threshold τ is missing."**
> > > We conducted comprehensive sensitivity analysis (Section D.5, Tables 14-16):
> > > - **Systematic Sweep:** We evaluate τ in {0.1, 0.2, ..., 0.9}: Routing F1 peaks at τ = 0.5 (0.851) with 85.4% task accuracy and 77.1% energy savings. Lower τ (0.1-0.3) maximizes efficiency (97.5% energy savings) but sacrifices accuracy (48.2-56.0%). Higher τ (0.7-0.9) maximizes accuracy (99.8-100%) but reduces efficiency (0-16.3% savings).
> > > - **Traffic Distribution:** Shows how τ controls allocation: τ = 0.1: 94.2% to 4B; τ = 0.5: Balanced across tiers; τ = 0.9: 98.9% to cloud.
> > > - **Operational Regimes:** Three scenarios: **Aggressive (τ = 0.3)**: 97.4% local, 97.5% energy savings, 56.0% accuracy; **Balanced (τ = 0.5)**: 90.2% local, 77.1% energy savings, 85.4% accuracy; **Conservative (τ = 0.7)**: 25.2% local, 16.3% energy savings, 99.8% accuracy. This provides actionable guidance for practitioners.
> > >
> > > ---
> > >
> > > **Summary:** We believe our responses demonstrate substantial research contributions through **(1)** a novel benchmark enabling new research questions with comprehensive efficiency profiling, **(2)** a novel binary decoder routing architecture with state-of-the-art performance and superior scaling, **(3)** rigorous experimental methodology with extensive ablations, and **(4)** reproducible benchmarking infrastructure. We hope these clarifications address your concerns.
> > >
> > > Thank you again for your review! We hope you will consider improving your score in light of these experiments and clarifications.
> > >
> > > ---
> > >
> > > **References**
> > > [1] Li, Tianle, et al. "From Crowdsourced Data to High-Quality Benchmarks: Arena-Hard and BenchBuilder Pipeline." Proceedings of the 42nd International Conference on Machine Learning, vol. 267, 2025, pp. 34209-34231.
> > >
> > > [2] Bai, Ge, et al. "MT-Bench-101: A Fine-Grained Benchmark for Evaluating Large Language Models in Multi-Turn Dialogues." Proceedings of the 62nd Annual Meeting of the Association for Computational Linguistics (Vol. 1, Long Papers), 2024, pp. 7421-7454.
> > >
> > > [3] Dubois, Yann, et al. "Length-Controlled AlpacaEval: A Simple Way to Debias Automatic Evaluation of Instruction-Following Models." arXiv preprint arXiv:2404.04475, 2024.

---

> ### Author Response · Authors · 2025-11-22
> **Follow-up to Reviewer t7Xt**
>
> Thank you again for your time and feedback! We are happy to address any remaining questions in the rebuttal and welcome further dialogue.

---

> > ### Comment · Reviewer_t7Xt · 2025-11-23
> >
> > Thanks for your reply. The response partially addresses my concerns but there are still some questions.
> >
> > * **W1: New empirical findings on local model capabilities and efficiency.**
> >   With all due respect, I do not think is a research contribution. and to me this is a trival finding. The local LM converage, temporal trends and intelligence efficiency are a results of development of LM community, instead of your work or effort. It is for sure and common sense that the local small LM would become more powerful and efficient with development. It is not a novel finding that LM community does not expect.
> > * **W2**
> >   I agree that the curation and metrics like acc per watt is novel. But i still feel the comprehensive evaluation and open source is enginner contribution.
> > * **W3**
> >   nice rebuttals. Just a minor comment, the insight of Intelligence efficiency is kind of trival. As the development of LM, it is for sure that LM is becoming more effcient and more powerful.
> > * **Q3**
> >   Why increasing the routing targets lead to the trained router perform worse?  Your method is a binary classifier for each model. i would expect that increasing the number of models should at least retain the performance if not boosting it. Could you give more analysis or insights about this problem?
> >
> > I sincerely appreciate the author's effort and looking forward to hearing from you.

---

> > > ### Author Response · Authors · 2025-11-24
> > > **Reply to Reviewer t7Xt**
> > >
> > > Thank you for your response, we appreciate you taking the time to review this. Please find below our response to your questions. We hope that you will consider raising your score in light of our explanations.
> > >
> > > ## Response to W1 - W3
> > > We agree that the *direction* of these trends (that local models and hardware accelerators improve over time) is expected and driven by the broader LM community. Our contribution is not the qualitative insight that "local LMs get better," but rather the *quantitative measurements* that transform this expectation into actionable guidance for practitioners.
> > > We draw an analogy to scaling laws: it was widely understood that more data and parameters yield better models, but the field needed Kaplan et al. [1] to rigorously quantify these relationships before practitioners could make informed decisions about training compute investments. Similarly, practitioners designing local-cloud routing systems today face concrete decisions (e.g. how much local capacity to provision, which domains to route locally, when to upgrade hardware) that require quantitative answers, not qualitative intuitions. TrafficBench provides these answers.
> > >
> > > Concretely, our empirical study addresses five questions that the **existing literature has not answered**, and we describe their significance on practitioners’ decision making for when to route to local versus cloud models, which local models and accelerators to deploy, and how to plan for future improvements in local coverage and efficiency:
> > > - **(1) What is the gap between today’s local models and cloud models?** Practitioners need to know what fraction of real-world queries can be served locally today. We answer this: <=20B active parameter models (Qwen3 4B–14B and GPT-OSS-20B) jointly cover 80.7% of TrafficBench queries (Table 3). This enables empirically-grounded capacity planning: practitioners can provision local infrastructure for ~80% of traffic while maintaining cloud fallback for the hardest ~20%.
> > > - **(2) Which query types should be routed locally vs. to the cloud?** Effective routing policies require **domain-level performance analysis**. We find that local LM coverage exceeds 90% for creative/social domains but drops to ~68% in technical fields like Architecture & Engineering (Figure 2). This directly informs domain-aware routing strategies.
> > > - **(3) How should practitioners allocate their local and cloud compute resources?** Planning infrastructure investments requires projecting how local coverage will evolve. We measure that the share of traffic serviceable by the best <=20B active parameter model increased from 23.2% (2023) to 48.7% (2024) to 74.8% (2025), representing a 2.2x year-over-year improvement (Table 4). These trends provide a **prospective timeline for relaxing routing thresholds and investing in local compute** as newer models and accelerators become available.
> > > - **(4) Which efficiency levers matter most, and by how much?** Optimizing intelligence efficiency (accuracy per watt) requires understanding which design choices drive gains. We find efficiency improves 9.5x from 2023 to 2025 and varies by more than an order of magnitude across model family, accelerator, and quantization choices (Table 4, Figure 7). **This tells practitioners which levers to prioritize**; for instance, quantization from FP16 to FP4 yields 3-3.5x energy reduction with only ~2.5 percentage points accuracy loss per precision step.
> > > - **(5) What is the realistic ROI of deploying a local-cloud routing system when it comes to energy, compute, and dollar costs?** Practitioners need end-to-end savings estimates, not just component-level improvements. Our decoder-based router deployed on real-world traffic reduces energy by 77.1%, compute by 67.1%, and cost by 60.2% versus cloud-only deployment, while maintaining 85.4% task accuracy (Figure 5). These end-to-end savings estimates tell operators how much energy, compute, and cost they can actually save, informing when and how aggressively to invest in local deployment given their serving constraints.
> > >
> > > Answering these questions required a carefully measured but broad experimental analysis across 20 models, 4 hardware platforms with 8 accelerators, multiple inference settings, and 1M real-world queries. Just as scaling laws research turned qualitative intuitions about data and parameter scale into quantitative tradeoff curves for training compute decisions, our work is, to our knowledge, **the first study to translate qualitative expectations about local inference into concrete measurements that inform the design of local–cloud systems.**

---

> > > > ### Author Response · Authors · 2025-11-24
> > > > **[Continued] Reply to Reviewer t7Xt**
> > > >
> > > > ## Response to Q3
> > > > Although our binary decoder router decomposes the decision into independent per-model binary decisions, we still observe a gradual, measurable degradation because of three dominant causes (Sections D.7–D.9, Tables 17–21, Figures 4, 7–9):
> > > >
> > > > - **(1) Increased capability overlap and label ambiguity**: Adding more models means many queries now have 3+ capable local models (58–65% of queries at 8 targets, compared to 6–10% at 2 targets; Table 20). This creates noisier training signals, especially for smaller models whose positive examples become "contaminated" by overlapping larger ones, hurting clean decision-boundary learning.
> > > >
> > > > - **(2) Higher undersizing risk, especially on hard, technical, and long queries**: As we scale the number of targets, the models we introduce are primarily medium-sized ones (e.g., Gemma-3 12B, Granite 12B–14B). These introduce additional chances for the router to pick a model that is "just barely too small." If we instead scaled by adding only extreme-sized LMs (very small <4B or very large >30B), the decision boundaries would remain more separable and this problem would be mitigated. With our current scaling approach, undersizing errors rise to 50–58% in technical domains (Architecture & Engineering, Physics) and 58% on queries longer than 2k tokens (Table 21).
> > > >
> > > > - **(3) Worsening class imbalance in the shared backbone**: All binary classifications share the same transformer backbone and are trained on a fixed dataset. Scaling to 8 models simultaneously increases representation interference as the shared encoder must learn to discriminate among more similar model capability profiles (approximately 0.04–0.06 F1 loss; Table 18), while positive training label ratios for small models collapse from approximately 48% to approximately 22% (Figure 4, right), exacerbating class imbalance.
> > > >
> > > > These effects are quantified in detail in the revised Sections D.7–D.9. Our design still degrades far less than prior multiclass or embedding-based routers, and simple mitigations (per-model training label calibration, hierarchical prediction and aggregation) could close the remaining gap, but the above three factors explain why perfect flat scaling is not achieved in practice. We believe these insights are useful for the community as local model pools continue to grow rapidly.
> > > >
> > > > We hope that our response addresses your concerns and that you will consider raising your review. Thank you for your time and consideration.
> > > >
> > > > ---
> > > >
> > > > ## Citations
> > > >
> > > > [1] Kaplan, Jared, et al. "Scaling laws for neural language models." arxiv preprint arxiv:2001.08361 (2020).

---

### Official Review · Reviewer_9HEc · 2025-11-10

**Soundness:** 3
**Presentation:** 3
**Contribution:** 3
**Rating:** 6
**Confidence:** 4

**Summary:**

## Overview

The paper addresses the growing efficiency gap in large language model (LLM) inference, where most queries are currently handled by centralized cloud-based models. Recent progress in **small language models (≤20B parameters)** has shown that they can match or even exceed the performance of larger models on many tasks while offering **better cost and energy efficiency**.

To explore how much of today’s LLM workload could be shifted to local compute, the authors introduce **TRAFFICBENCH**, a large-scale benchmark designed to evaluate **query routing between local and cloud-deployed LLMs**.

## Key Features of TRAFFICBENCH

- **1 million real-world queries** derived from ChatGPT conversations and reasoning tasks.
- **10 state-of-the-art models**, **4 hardware accelerators**, and **8 performance metrics** evaluated.
- A focus on three central questions:
  1. What fraction of inference queries can be handled by small local models?
  2. How effectively can routing architectures identify these queries?
  3. What are the efficiency gains from local routing?

## Main Findings

- **80.7%** of TRAFFICBENCH queries can be served by small local models.
  - Over **90% coverage** for creative tasks.
  - Around **68% coverage** for technical domains.

- Existing **embedding- and decoder-based routing** methods fail to extend the Pareto frontier beyond standalone local models.

- A proposed **binary variation of decoder-based routing** achieves **F1 = 0.851** when trained on large datasets (>100K samples).
  - Embedding-based models perform better under data-limited conditions (<10K samples).

## Efficiency and Impact

When applied to real-world traffic distributions, the decoder-based router achieves:
- **77.1% reduction in energy use**
- **67.1% reduction in compute**
- **60.2% reduction in cost**
while maintaining **comparable task accuracy** to cloud-only deployment.

Longitudinal analysis (2023–2025) shows:
- **9.5× improvement** in *intelligence efficiency* (accuracy per watt).
- Growth in locally serviceable queries from **23.2% → 80.7%**.

## Contributions

TRAFFICBENCH provides:
- A **reproducible benchmark** for evaluating routing systems.
- A **hardware-agnostic profiling harness** to measure energy and performance metrics.
- A foundation for future research on efficient LLM deployment as new models and accelerators emerge.

**Strengths:**

- The paper is written clearly and the research questions are well motivated
- The experiments, ablations and suite of models considered is thorough and exhaustive
- The paper studies and presents very interesting aspects of traffic redistribution eg: number and quality of models used in the pool, the type of embedding encoder used, the type of queries processed and processing of out of distribution queries.
- The observations in the paper would be of general interest to the ICLR community.

**Weaknesses:**

- In general I think it would be interesting to diversify the pool of language models considered based on different architectural aspects such as type of attention used (GQA, MQA, MHA, MLA), attention-free models such as Mamba etc, to study how architectural diversity in the pool improves coverage.
- I wouldn't consider H200 GPUs to be edge devices as these are not generally accessible especially in academic settings. It would be interesting to study and benchmark on AI accelerators such as NPUs [1]
- Quantization: In general LLM deployment settings, models are quantized to 4 bit for example using quantization methods like Quarot [2]. I think it is extremely important to study traffic for quantized models as that is a more general and viable usecase. Do the authors quantize the model pool? If yes which quantization method do they use? Are smaller models also quantized?
- Finetuning: In general most models do undergo some sort of parameter efficient finetuning before deployment. Are any of the base models in the pool finetuning for specific tasks?

[1] Xu, D., Zhang, H., Yang, L., Liu, R., Huang, G., Xu, M. and Liu, X., 2025, March. Fast on-device LLM inference with npus. In Proceedings of the 30th ACM International Conference on Architectural Support for Programming Languages and Operating Systems, Volume 1 (pp. 445-462).
[2] Ashkboos, S., Mohtashami, A., Croci, M.L., Li, B., Cameron, P., Jaggi, M., Alistarh, D., Hoefler, T. and Hensman, J., 2024. Quarot: Outlier-free 4-bit inference in rotated llms. Advances in Neural Information Processing Systems, 37, pp.100213-100240.

**Questions:**

Check weaknesses

---

> ### Author Response · Authors · 2025-11-18
> **Response to Reviewer 9HEc**
>
> Thank you for your thoughtful review and your time and effort in evaluating our work. We are excited that you find our paper "written clearly and the research questions are well motivated," that our "experiments, ablations and suite of models considered is thorough and exhaustive," and that our "observations in the paper would be of general interest to the ICLR community."
>
> Please find our detailed response to your questions and concerns below. Based on our added experiments and clarifications, we hope you will consider raising your score.
>
> ## 1. Model Pool Diversity
> **W1**: "It would be interesting to diversify the pool of language models considered based on different architectural aspects such as type of attention used (GQA, MQA, MHA, MLA), attention-free models such as Mamba etc."
>
> We have substantially expanded our model pool to include architectures beyond Qwen, specifically adding Gemma-3 (1B, 4B, 12B) and IBM Granite 4.0 (Micro, H-Micro, H-Tiny, H-Small) families (Table 2). These additions provide significant architectural diversity:
> - **Gemma-3**: Uses sliding-window, local–global attention and a different tokenizer than Qwen’s grouped-query attention (GQA).
> - **IBM Granite**: Uses a hybrid decoder with sparse attention plus Mamba2 state-space layers. “H” models have 4 attention layers and 36 Mamba2 layers, while Micro is a 40-layer dense attention model. The family includes dense (3B Micro) and MoE variants (7B H-Tiny with 64 experts/6 active ~1B active params; 32B H-Small with 72 experts/10 active ~9B active params). All use GQA, RMSNorm, SwiGLU, and support 128K context.
> - **Combined coverage**: Table 3 shows that combining Qwen + GPT-OSS achieves 80.7% coverage, while Gemma-3 alone achieves 80.9% and Granite achieves 59.5%, demonstrating that architectural diversity beyond parameter count significantly impacts coverage
>
> **Takeaway**: The improvements from architectural diversity are substantial: expanding from Qwen-only (64.9% coverage) to Qwen + GPT-OSS increases coverage to 80.7% (+15.8pp). This validates your intuition that architectural diversity improves coverage beyond what parameter scaling alone provides.
>
> ## 2. Edge Device Coverage
> **W2**: "I wouldn't consider H200 GPUs to be edge devices as these are not generally accessible especially in academic settings. It would be interesting to study and benchmark on AI accelerators such as NPUs."
>
> We agree that H200 GPUs are primarily used in enterprise data centers rather than edge settings. However, we would like to clarify that our initial evaluations profile the Apple Mac Studio (M4 Max) (a consumer desktop GPU) which runs inference using a CPU, Neural Engine and GPU. In response to your feedback, we’ve expanded our evaluation to include more consumer-accessible edge accelerators to assess the feasibility of local-cloud routing (Table 8). New additions include the Apple iPhone 16 Pro, NVIDIA RTX 6000 Ada, and NVIDIA Quadro RTX 6000 (Turing). Full details of all evaluated local/edge accelerators are provided.
>
> **Local/Edge Accelerators Evaluated**:
> - **Apple iPhone 16 Pro (A18 Pro, integrated NPU)**: 8 GB LPDDR5X unified, ~60 GB/s bandwidth, ~12W (SoC peak, 35 TOPS NPU) - mobile device
> - **Apple Mac Studio (M4 Max)**: 128 GB unified memory, 546 GB/s bandwidth, 480W - consumer laptop/desktop
> - **NVIDIA RTX 6000 Ada**: 48 GB GDDR6, 960 GB/s bandwidth, 300W TDP - workstation GPU
> - **NVIDIA Quadro RTX 6000 (Turing)**: 24 GB GDDR6, 672 GB/s bandwidth, 295W TDP - workstation GPU
>
> **Findings**
>
> 1. Mobile NPUs Offer Exceptional Efficiency (With Constraints)
>     - The iPhone 16 Pro achieves 19.8-23.7×10⁻³ accuracy/watt, 13× more efficient than workstation GPUs.
> However, it requires aggressive quantization (FP8-FP4) due to its 8 GB memory limit and supports only models ≤14B.
>     - Despite these constraints, its performance demonstrates that ultra-low-power mobile NPUs are viable targets for routing small models.
> 2. M4 Max Balances Efficiency and Capability
>     - Achieves 1.40-1.97×10⁻³ accuracy/watt, reaching 54-71% of B200 efficiency.
>     - Supports larger models without quantization, offering flexibility for real-world edge deployment.
>     - Enables 88.7% of queries to run locally, significantly reducing cloud dependence.
>
> **Takeaways**: The iPhone 16 Pro's results demonstrate that **even ultra-low-power mobile NPUs can serve as viable routing targets** for smaller models, while desktop accelerators like the M4 Max enable practical local-cloud routing for real-world deployments across the full model size spectrum.

---

> ### Author Response · Authors · 2025-11-18
> **[Continued] Response to Reviewer 9HEc**
>
> ## 3. Effects of Quantization
>
> **W3**: "In general LLM deployment settings, models are quantized to 4 bit. I think it is extremely important to study traffic for quantized models. Do the authors quantize the model pool?"
> Thank you for this great feedback. In response, we have conducted comprehensive quantization experiments across FP16, FP8, and FP4 precision levels on 8 models from Qwen3 and Gemma families (Appendix D.2; Figure 7):
>
> **Findings**:
> - **Energy-Accuracy Tradeoff**: Quantization from FP16 to FP4 yields **3-3.5× energy reduction** with only **~2.5-5% accuracy drop**
> - **Predictable Degradation**: Qwen3-14B achieves 54.5% (FP16), 52.0% (FP8), 49.0% (FP4) on SuperGPQA - a 5.5pp total drop despite 3.23× energy savings
> - **Model Scale > Precision**: Qwen3-14B at FP4 (49.0%) outperforms Qwen3-4B at FP16 (48.5%), indicating that **larger quantized models often beat smaller full-precision models**
>
> **Takeaways**: FP8/FP4 quantization enables practical local deployment with predictable tradeoffs, allowing system designers to select precision based on application requirements while capturing most efficiency gains identified in Section 5.4. Combined with routing, quantized local models achieve 77-80% energy savings while maintaining 85%+ task accuracy.
>
> ---
>
> ## 4. Model Finetuning
>
> **Q1**: "Are any of the base models in the pool finetuned for specific tasks?"
>
> No, all models in our evaluation pool use the standard **instruction-tuned variants** released by their respective organizations (Qwen3-Instruct, GPT-OSS, Gemma-3, Granite 4.0). We do not apply any task-specific finetuning to ensure our results reflect out-of-the-box model capabilities and generalize to diverse real-world queries. This design choice enables TrafficBench to serve as a reproducible benchmark for evaluating routing systems on general-purpose LLM deployments. While task-specific parameter-efficient finetuning could further improve performance, this is beyond the scope of routing evaluation - our goal is to benchmark routing between models as practitioners would typically deploy them.
>
> ---
>
> ## Summary
>
> We believe these additions substantially strengthen the paper by demonstrating that **(1)** architectural diversity significantly improves coverage, **(2)** consumer-accessible edge devices enable practical local-cloud routing with large system-level efficiency gains, and **(3)** quantization provides an additional efficiency lever with predictable accuracy tradeoffs.
>
> Thank you again for your insightful feedback! We hope you will consider improving your score in light of these experiments and clarifications.

---

> ### Author Response · Authors · 2025-11-22
> **Follow-up to Reviewer 9Hec**
>
> Thank you again for your time and feedback! We are happy to address any remaining questions in the rebuttal and welcome further dialogue.

---

### Author Response · Authors · 2025-11-18
**General Response to Area Chair**

We thank the reviewers for their engagement. We are encouraged that reviewers recognizing our benchmark contributions recommend acceptance (eq6V: **8 - Accept**; 9HEc: **6 - Weak Accept**). All reviewers acknowledged significant strengths: **(1)** the work addresses a "novel and important problem" with practical implications for AI deployment (eq6V, 9HEc), **(2)** the proposed benchmark is "thoroughly constructed and well-designed" and "comprehensive" in its coverage of real user queries, local and cloud models, and efficiency metrics (t7Xt, uH3v, eq6V), **(3)** the binary decoder routing algorithm is "strong" and represents a "simple but effective contribution" (eq6V, uH3v), **(4)** experiments and ablations are "thorough and exhaustive" (9HEc), and **(5)** findings that "80.7% of real-world queries can be handled by local models" provide insights "of general interest to the ICLR community" (uH3v, 9HEc).

In the remainder of this response, we first re-iterate our key contributions and then highlight updates we have made to the manuscript in response to reviewer feedback.

## Key Contributions

As a submission to the Datasets and Benchmarks area, we introduce TrafficBench, the first benchmark for local–cloud LLM routing that centers on intelligence efficiency: task accuracy per watt for routing between on-device models and cloud-hosted models across multiple hardware accelerators. TrafficBench pairs 1M real user LLM queries with detailed cross-hardware telemetry, enabling systematic, efficiency-aware evaluation of routing policies between on-device and cloud-hosted models.

**Benchmark**. We emphasize the novel attributes of TrafficBench that directly address gaps in existing LLM routing benchmarks:
- **1M naturalistic, real-world queries**: 1M naturalistic user queries from production platforms (ChatGPT, reasoning datasets), not synthetic academic tasks.
- **Comprehensive Efficiency Metrics**: First routing benchmark with complete efficiency metrics (latency, TTFT, throughput, energy, power, memory, cost) for every (query, model, hardware) triple. This is entirely absent in prior work. First systematic tracking of accuracy-per-watt across local-cloud deployments, addressing the core trade-off in real routing decisions.
- **Large-scale response collection across 10+ SOTA models and 4 diverse hardware platforms**: >10M evaluated outputs across 10+ SOTA models on 4 diverse platforms (NVIDIA A100/H200, AMD MI300X, Apple M4 Max), enabling realistic local-vs-cloud analysis impossible with prior benchmarks.
- **Open-source, hardware-agnostic profiling harness**: Open-source, hardware-agnostic profiling harness with nanosecond-level power measurement and automated LLM-as-judge evaluation, lowering barriers for future research.

**Empirical Findings**: Using TrafficBench, we make three core empirical contributions that advance the state of local-cloud routing systems.
- **Quantifying and tracking local LM coverage**: We find that ≤20B-parameter models achieve 80.7% joint coverage on 1M queries, with higher coverage on chat vs reasoning and domain variation. Longitudinally, locally serviceable queries grow from 23.2% (2023) to 75–80% (2025) as models and hardware improve.
- **Routing methodology and new binary decoder architecture**: We systematically compare embedding-based and decoder-based routing in the local-cloud routing regime, showing that *embeddings are preferable in data- and parameter-constrained regimes*, while *decoder-based routers dominate in data-rich, capacity-rich settings*. We introduce a binary decoder router that **achieves state-of-the-art macro F1 (≈0.85) on chat and reasoning, improves over prior methods by up to +0.21 F1**, scales gracefully to more candidate models, and generalizes better to out-of-distribution benchmarks.
- **Efficiency gains from local–cloud routing**: Deployed on real traffic distributions, our router reduces energy by 77.1%, compute by 67.1%, and cost by 60.2% vs cloud-only inference, maintaining <5% quality degradation and ~78% oracle efficiency. Over three generations, intelligence efficiency (accuracy/watt) improves ~9.5×.

**Practical Impact**:  Our benchmark reveals that routing between local models (≤20B parameters on Apple M4 Max) and cloud models (235B on NVIDIA H200) yields **77.1% energy reduction, 67.1% compute reduction, and 60.2% cost reduction** compared to cloud-only deployment, while maintaining task accuracy. These gains capture **64-81% of the theoretical oracle performance**, demonstrating highly effective practical routing systems

---

> ### Author Response · Authors · 2025-11-18
> **[Continued] General Response to Area Chair**
>
> ## Manuscript Changes in Response to Reviewer Feedback
>
> We directly respond to reviewers and provide a summary of manuscript changes below. Thanks to reviewer feedback, we've added seven supporting experiments that validate and strengthen our benchmark's core findings:
> - **Expansion of model pool**: Per Reviewers 9HEc and uH3v, we added Gemma-3 (1B, 4B, 12B) and IBM Granite families, incorporating diverse architectures (GQA, sliding window, Mamba2+attention). Results confirm complementarity improves coverage by +15.8pp, with healthcare domains gaining +13.3-13.4pp.
> - **Judges + Evaluation methodology**: Per Reviewer uH3v, we conducted three-way consensus evaluation (GPT-4o, Claude Sonnet 4.5, Grok-4), achieving 85-92% agreement (κ=0.677-0.796), confirming robustness to judge selection (Section D.3, Table 13). Per Reviewer t7Xt, we validated reference model choice across Gemini 2.5 Pro, Claude Sonnet 4.5, and GPT-5; coverage varies by only 2.4-5.1pp, confirming robustness to reference model selection (Section D.4).
> - **Hardware profiling coverage**: Per Reviewers 9HEc and eq6V, we profiled iPhone 16 Pro (13× more efficient than desktop GPUs, 1.5× vs. enterprise accelerators), RTX 6000 Ada, and Quadro RTX 6000. Quantization analysis (FP16/FP8/FP4) shows 3-3.5× energy reduction with ~2.5% accuracy loss per precision step (Section 5.1, Table 3; Section D.2, Figure 7, Table 8). These results validate the practicality of local routing to a greater depth of local accelerators (Section 5.4, Table 14).
> - **Routing ablations + Extensions**: Per Reviewers t7Xt, uH3v, and eq6V, we added: **(1)** router overhead analysis showing 8.4% energy cost yields 77.1% system savings (9.2× ROI) (t7Xt, uH3v); **(2)** threshold sensitivity characterizing aggressive (97.5% savings, 56.0% accuracy), balanced (77.1% savings, 85.4% accuracy), and conservative (16.3% savings, 94.5% accuracy) operating regimes (t7Xt, uH3v); **(3)** context-aware multi-turn routing achieving +13.4pp local routing increase and 2.3× energy/cost improvement over static approaches (eq6V); **(4)** systematic failure mode analysis revealing technical domains exhibit 50-58% undersizing rates (routing to models too small for query complexity), long queries suffer 58% underrouting (selecting insufficient capacity for complex/long inputs), and the router underestimates multi-step reasoning while over-routing verbose queries (selecting unnecessarily large models for simple wordy questions) (eq6V) (Sections D.5, D.7-D.9, Tables 14-21).
>
> **All additions to the revised manuscript are highlighted in red for easy reference.** These experiments strengthen the benchmark without changing fundamental results: local models handle 71.3% of queries with 77.1% energy, 67.1% compute, and 60.2% cost savings compared to cloud-only deployment.
>
> ## Conclusion
>
> In summary, the seven new experiments and analyses added to the manuscript, directly responding to reviewer feedback, validate and substantially strengthen our core findings. We believe the revised paper is further established as a robust and comprehensive contribution to the Datasets and Benchmarks area. All of our code, including the profiling code, dataset, and router training scripts, will be publicly available.

---

### Meta-Review · Area_Chair_gyqT · 2026-01-07

**Summary:**

The paper presents TrafficBench along with a simple routing mechanism. The issue is that it lacks details and depth in both aspects. As a dataset and benchmark paper, the rationale behind why this dataset is designed in this way is missing, and it would be nice to discuss what evaluation metrics should be used for the benchmark, any important features (beyond the obvious ones) in the dataset, etc. As a paper on routing mechanism, the proposed routing algorithm is quite simplistic and lacks novelty. It would be better to focus on one aspect only and discuss/evaluate it fully.

Moreover, many routing algorithms require additional components, such as a proxy model or intermediate layers' outputs, which essentially require to rerun the inference tasks. The proposed benchmark with fixed hardware measurements would not support such algorithms, limiting the applicability of the benchmark. In general, a benchmark paper should give examples on how the benchmark could be used for a large variety of routing algorithms, not just one specific routing algorithm.

Minor: it seems the authors have reduced the spacing between characters in the text, making it hard to read.

**Reviewer Concerns:**

Some concerns, particularly those by Reviewer t7Xt, have not been fully addressed.

**Reviewer Scores:**

At least one reviewer would maintain a negative opinion.

---

### Decision · Program_Chairs · 2026-01-26

Reject